# Telomere length regulation by Rif1 protein from *Hansenula polymorpha*

**Alexander N Malyavko[1]\*, Olga A Petrova[1], Maria I Zvereva[1], Vladimir I Polshakov[2], Olga A Dontsova[1,3,4]**

[1]Faculty of Chemistry and Belozersky Institute of Physico-Chemical Biology, Lomonosov Moscow State University, Moscow, Russian Federation; [2]Center for Magnetic Tomography and Spectroscopy, Faculty of Fundamental Medicine, Lomonosov Moscow State University, Moscow, Russian Federation; [3]Center of Life Sciences, Skolkovo Institute of Science and Technology, Moscow, Russian Federation; [4]Shemyakin-Ovchinnikov Institute of Bioorganic Chemistry of the Russian Academy of Sciences, Moscow, Russian Federation

**Abstract** Rif1 is a large multifaceted protein involved in various processes of DNA metabolism – from telomere length regulation and replication to double-strand break repair. The mechanistic details of its action, however, are often poorly understood. Here, we report functional characterization of the Rif1 homologue from methylotrophic thermotolerant budding yeast *Hansenula polymorpha* DL-1. We show that, similar to other yeast species, *H. polymorpha* Rif1 suppresses telomerase-dependent telomere elongation. We uncover two novel modes of Rif1 recruitment at *H. polymorpha* telomeres: via direct DNA binding and through the association with the Ku heterodimer. Both of these modes (at least partially) require the intrinsically disordered N-terminal extension – a region of the protein present exclusively in yeast species. We also demonstrate that Rif1 binds Stn1 and promotes its accumulation at telomeres in *H. polymorpha*.

**\*For correspondence:**
malyavkoan@gmail.com

**Competing interest:** The authors declare that no competing interests exist.

## Editor's evaluation

This paper presents a biochemical and functional analysis of the telomere-binding protein Rif1 (Rap1-interacting factor 1) in a budding yeast species (Hansenulapolymorpha; Hp) distantly related to the well-studied *Saccharomyces cerevisiae*. They present convincing evidence for a DNA-binding function of HpRif1 encoded in a conserved N-terminal domain predicted to be intrinsically disordered, as well as a region in the N-terminus that promotes interaction with the Hp Ku heterodimer. They also identify and characterize a functional interaction of Hp Rif1 with the Stn1 protein. Given these findings and the apparent absence of strong interaction with Rap1 in Hp, the authors provide an emerging picture of Hp Rif1 function that differs from that seen in *Saccharomyces cerevisiae*, consistent with the general picture of the rapid evolution of telomeric proteins.

## Introduction

Chromosomes of most eukaryotic organisms end in distinctive nucleoprotein structures, called telomeres, which are essential for protection of chromosomes from degradation and fusions by the DNA repair machinery (*de Lange, 2018*). Short G/C-rich telomeric DNA repeats provide a platform for loading of a specific set of telomeric proteins forming complex dynamic assemblies at the ends of the chromosomes. Dividing cells are constantly losing telomeric repeats due to incomplete replication unless counteracted by either recombination or activity of the ribonucleoprotein complex telomerase. Replenishing of telomeric DNA by telomerase is crucial for long-term proliferation of the majority of

eukaryotic cell types. This process is highly coordinated, and plenty of positive and negative regulators of telomerase have been identified to date, many of which are components of the telomeric chromatin (*Lee et al., 2021*; *Lim and Cech, 2021*; *Shay and Wright, 2019*).

In budding yeast *Saccharomyces cerevisiae*, the double-stranded portion of telomeres is bound by Rap1 through its duplicated MYB domain (*Henry et al., 1990*; *Shore and Nasmyth, 1987*). Two additional telomeric proteins – Rif1 and Rif2 – are recruited to telomeres via interaction with Rap1 C-terminus (RCT) (*Hardy et al., 1992*; *Wotton and Shore, 1997*). Both of them restrict telomerase action and their double deletion (as well as deletion of the RCT) leads to telomere hyperelongation (*Teng et al., 2000*; *Wotton and Shore, 1997*). Rap1 also recruits the histone deacetylase complex Sir2/Sir3/Sir4 (through an interaction between Sir4 and RCT) providing the basis for the silencing of telomere-proximal genes (*Moretti et al., 1994*). The C-terminal portion of Rif1 contains a Rap1-binding motif (Rif1$_{RBM}$) and a tetramerization module (Rif1$_{CTD}$), while Rif2 contains two Rap1-binding sites, therefore Rap1/Rif1/Rif2 can form complex DNA-protein structures at the chromosome ends, inhibiting both telomerase recruitment and telomere silencing (*Shi et al., 2013*). In addition, Rif1 can localize at telomeres in a Rap1-independent fashion through its large HEAT repeats containing N-terminal domain (Rif1$_{NTD}$) that exhibits strong affinity to DNA with some preference towards 3'-overhang containing ss-ds junctions (*Mattarocci et al., 2017*). This binding mode is important for telomerase regulation and attenuation of end resection at telomeres. Recently, the ScRif1$_{NTD}$ was proposed to contain an interaction site for a yet unidentified protein partner at telomeres (*Shubin et al., 2021*).

The single-stranded region of telomeres (3'-overhang) is bound and protected by the Cdc13 protein (*Garvik et al., 1995*; *Lin and Zakian, 1996*; *Nugent et al., 1996*) together with its partners Stn1 and Ten1 (the CST complex; *Grandin et al., 2001*; *Grandin et al., 1997*; *Mersaoui and Wellinger, 2019*; *Wellinger and Zakian, 2012*). Cdc13 is a central hub for telomeric DNA synthesis as it mediates the main pathway of telomerase recruitment (synthesis of the G-strand), as well as assists in loading of the Polα(synthesis of the C-strand) (*Qi and Zakian, 2000*; *Wellinger and Zakian, 2012*). Telomerase recruitment in *S. cerevisiae* also depends on another telomeric complex – the Ku70/Ku80 heterodimer (Ku): it binds a stem-loop structure within telomerase RNA and the Sir4 protein at telomeres (*Hass and Zappulla, 2015*; *Peterson et al., 2001*; *Stellwagen et al., 2003*). In addition, Ku can bind DNA directly and apparently does so at sub-telomeric regions and, perhaps, at the ds-ss junction of the telomere (*Larcher et al., 2016*; *Lopez et al., 2011*). Ku loss leads to telomere shortening and Exo1-dependent accumulation of single-stranded telomeric DNA (*Bonetti et al., 2010*; *Boulton and Jackson, 1996*; *Gravel et al., 1998*; *Polotnianka et al., 1998*; *Porter et al., 1996*).

Almost every telomeric chromatin component plays important roles outside telomeres. Rap1 protein is a central transcription factor, controlling expression of hundreds of genes in budding yeast (*Azad and Tomar, 2016*). A regulatory role during RNA PolII transcription was recently described for ScCST complex (*Calvo et al., 2019*). Ku heterodimer is a key component of the non-homologous end joining (NHEJ) machinery (*Frit et al., 2019*). Rif1 protein binds Glc7 (PP1) phosphatase and this interaction is important for regulation of replication initiation at several origins (*Davé et al., 2014*; *Hiraga et al., 2014*; *Mattarocci et al., 2014*), but not for telomere length control (*Shubin and Greider, 2020*). Interestingly, this function in replication, but not the telomeric role, is conserved among eukaryotic Rif1 homologues (see *Alavi et al., 2021* for a recent review). The only known species outside budding yeast clade which contains Rif1 as a component of 'normal' telomeric chromatin is fission yeast *Schizosaccharomyces pombe*, although SpRif1 is recruited through an interaction with Taz1 protein rather than Rap1 (*Kanoh and Ishikawa, 2001*). Mammalian Rif1 associates with dysfunctional telomeres (and other types of DSB) together with 53BP1, limiting accumulation of ssDNA through a mechanism involving Rev7, the Shieldin complex, CST and Polα, and driving the DSB repair toward NHEJ (*Barazas et al., 2018*; *Chapman et al., 2013*; *Dev et al., 2018*; *Di Virgilio et al., 2013*; *Escribano-Díaz et al., 2013*; *Gupta et al., 2018*; *Mirman et al., 2018*; *Noordermeer et al., 2018*; *Tomida et al., 2018*; *Zimmermann et al., 2013*). Interestingly, *S. cerevisiae* Rif1 also localizes to non-telomeric DSBs and promotes NHEJ through its DNA binding NTD, suggesting that Rif1 may be a conserved NHEJ factor (*Mattarocci et al., 2017*).

The thermotolerant methylotrophic budding yeast species *Hansenula polymorpha* DL-1 is distantly related to *S. cerevisiae*. Several interesting differences in telomere biology between the two species have already been documented. Rap1 has two paralogues in *H. polymorpha* (Rap1A and Rap1B) with distinct DNA recognition properties (*Malyavko et al., 2019*). HpRap1A is located at the subtelomeric

regions with no reported telomeric role. HpRap1B is the major telomeric dsDNA binder with the ability to control telomere length, although the primary target of its inhibition appears to be recombination rather than telomerase (*Malyavko et al., 2019*). The *RIF2* gene is absent from *H. polymorpha* genome (and other yeasts outside *Saccharomycetaceae* family). A shorter version of Cdc13 protein is present in *H. polymorpha*, which possesses strong affinity for telomeric G-strand in vitro and binds Stn1 protein (*Malyavko and Dontsova, 2020*). HpCdc13 also interacts with HpTERT in the yeast-two-hybrid (Y2H) assay.

Here, we explored the conservation of the telomeric roles of Rif1 protein within budding yeast clade by investigating Rif1 function in the telomere maintenance of *Hansenula polymorpha*. We found that Rif1 restricts telomerase action at *H. polymorpha* telomeres. We demonstrate that the Rif1 N-terminal extension (Rif1$_{NTE}$) – a portion of Rif1 present exclusively in yeast homologues of the protein – is an intrinsically disordered domain with the ability to bind DNA in vitro. We show that this DNA-binding domain helps to recruit Rif1 to telomeres. Moreover, we found that the interaction with the Ku heterodimer is required for Rif1 telomeric localization in *H. polymorpha*. Finally, we found that the conserved N-terminal domain of HpRif1 binds to HpStn1 protein, and this interaction is crucial for HpStn1 recruitment at telomeres.

## Results

### Rif1 protein regulates telomere length in *H. polymorpha*

Iterative PSI-BLAST search against NCBI protein database yielded a single plausible *H. polymorpha* Rif1 homologue: the open reading frame HPODL_04218 (UniProt accession number W1QFB8). Amino acid sequence similarity between ScRif1 and HpRif1 is very low (16% identity); however, alignment of multiple budding yeast homologues revealed several conserved elements within HpRif1 (*Figure 1A*, *Supplementary file 1*): the RVxF/SILK motif (residues 277–306), the N-terminal HEAT repeats containing domain (Rif1$_{NTD}$, residues ~ 310–1000) and the C-terminal domain (Rif1$_{CTD}$, residues ~ 1475–1521). HpRif1 also contains large potentially disordered regions (residues ~ 1000–1475 and 1–277). The residues located N-terminally to the RVxF/SILK motif we term N-terminal extension or NTE (residues 1–277).

To test whether the identified HpRif1 homologue is involved in telomere maintenance, we deleted the *RIF1* gene and measured telomere length in the mutant strain. We observed markedly elongated telomeres in the *Δrif1* strain (*Figure 1B*). Concomitant deletion of the *RAD52* gene did not influence telomere overelongation in the *Δrif1* strain, whereas deletion of telomerase RNA gene (*TER*) led to telomere shortening (*Figure 1B*). The *Δrif1Δter* double knock-out strain senesced upon continuous propagation, with no substantial increase in survival compared to the *Δter* strain (*Figure 1C*). These observations suggest that the more likely target of Rif1's inhibition is telomerase, and not the recombination pathway of telomere elongation.

Next, we investigated telomeric localization of HpRif1. We created the *RIF1-HA* strain expressing HpRif1 protein tagged with hemagglutinin epitope (HA) at the C-terminus. Chromatin immunoprecipitated from this strain via anti-HA antibodies was enriched with telomere proximal fragment 'TEL' compared to the internal fragment 'ALA1', suggesting that Rif1 localizes at *H. polymorpha* telomeres in vivo (*Figure 1D*). Thus, Rif1 is a telomeric protein inhibiting telomerase action in *H. polymorpha* similar to its *S. cerevisiae* homologue.

### Involvement of Rap1 in recruitment of Rif1 to telomeres

Telomeric localization of ScRif1 is largely dependent on the interaction with ScRap1, and mutation in the Rif1$_{RBM}$, located within the C-terminal part of the protein, greatly reduce the amount of telomere-bound ScRif1 (*Shi et al., 2013*). We could not find a similar Rap1-binding motif in the HpRif1 sequence (*Supplementary file 1*), suggesting that HpRif1 may be recruited at telomeres in a Rap1-independent fashion, or such RBM is located in a different region of the protein. Phenotypic consequences of the ScRif1 loss or removal of the C-terminal domain of ScRap1 are similar, as they both result in the telomerase-dependent telomere overelongation. Telomere elongation in the *Δrif1* strain of *H. polymorpha* appear to rely exclusively on telomerase action (*Figure 1B and C*). However, we have previously found that elongation of telomeres in the B$^{1–526}$ strain (expressing C-terminally truncated Rap1B) is largely Rad52-dependent, providing further indication for a Rap1-independent Rif1 function. If Rap1B

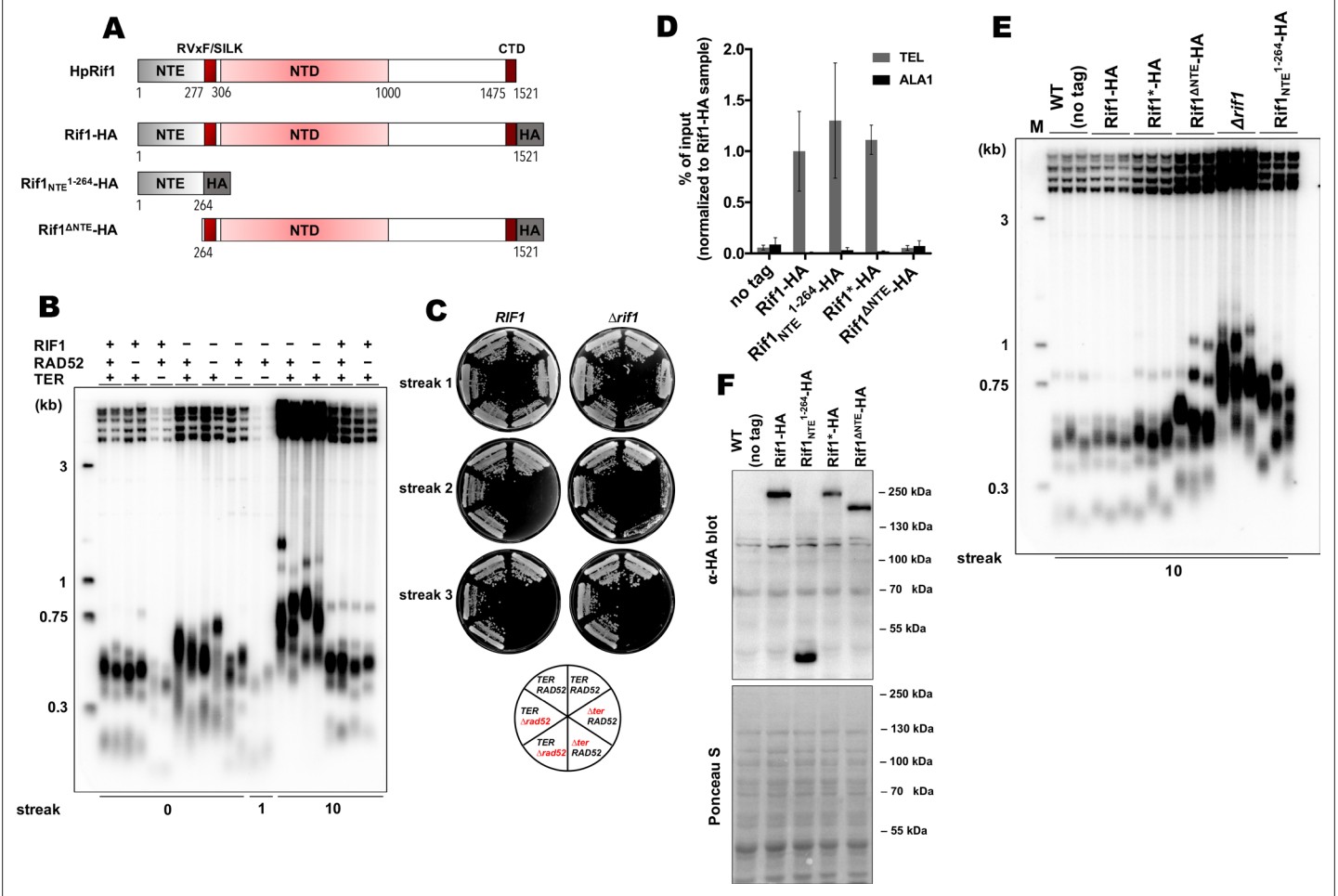

**Figure 1.** Rif1 regulates telomere length in *H. polymorpha*. (**A**) Schematic illustration of the domain organization of the full length HpRif1 and HA-tagged Rif1 fragments expressed in *H. polymorpha*. (**B**) Southern blot analysis of terminal restriction fragments from the indicated mutant strains. Genomic DNA was isolated from the strains after the Nth streak (where N is a number under a lane; each streak is ~20 generations). '0' streak – gDNA was isolated from the colonies on the transformation plate, without additional restreaks. 'M' – telomeric DNA containing fragments that served as markers of length (their sizes are indicated on the left of each blot). (**C**) Viability of the strains with the indicated genotypes was monitored during three serial restreaks on YPD agar plates, the plates were photographed after 2 days growth at 37 °C. (**D**) ChIP analysis. Chromatin from the indicated strains was immunoprecipitated on anti-HA magnetic beads. DNA was analyzed by qPCR with primers targeting either subtelomere region of the right end of chromosome VII ('TEL') or ALA1 gene locus (negative control, 'ALA1'). The amount of DNA fragments in the IP samples as a percentage of the input DNA was calculated, the % of input of the 'TEL' Rif1-HA sample was set to 1. Error bars indicate SD, n = 3. 'Rif1*-HA' – the strain expressing the Rif1-HA protein but in a slightly different background (see Materials and methods). (**E**) Same as (**B**) but with different strains. (**F**) Western blot analysis of the total proteins isolated from the indicated strains using antibodies targeting HA epitope (upper panel). Ponceau S-stained membrane (lower panel) served as a loading control.

The online version of this article includes the following source data and figure supplement(s) for figure 1:

**Source data 1.** Numerical data used to generate *Figure 1*.

**Source data 2.** The original (raw unedited) gels/blots for *Figure 1*.

**Source data 3.** The original (raw unedited) gels/blots for *Figure 1*.

**Figure supplement 1.** Binding of Rap1B and Rif1 at the internal telomere locus.

**Figure supplement 1—source data 1.** Numerical data used to generate *Figure 1—figure supplement 1*.

is not involved in Rif1 recruitment – then Rif1 should not bind to double stranded telomeric DNA. To test this, we constructed the intTEL18 strain of *H. polymorpha* by inserting 18 telomeric repeats upstream of the WSC3 gene on the chromosome I (*Figure 1—figure supplement 1A*). As expected, Rap1B binds telomeric repeats with similar efficiency at the internal locus or at the chromosome end (only ~2 fold difference between WSC3 and TEL signals for the intTEL18 strain, *Figure 1—figure*

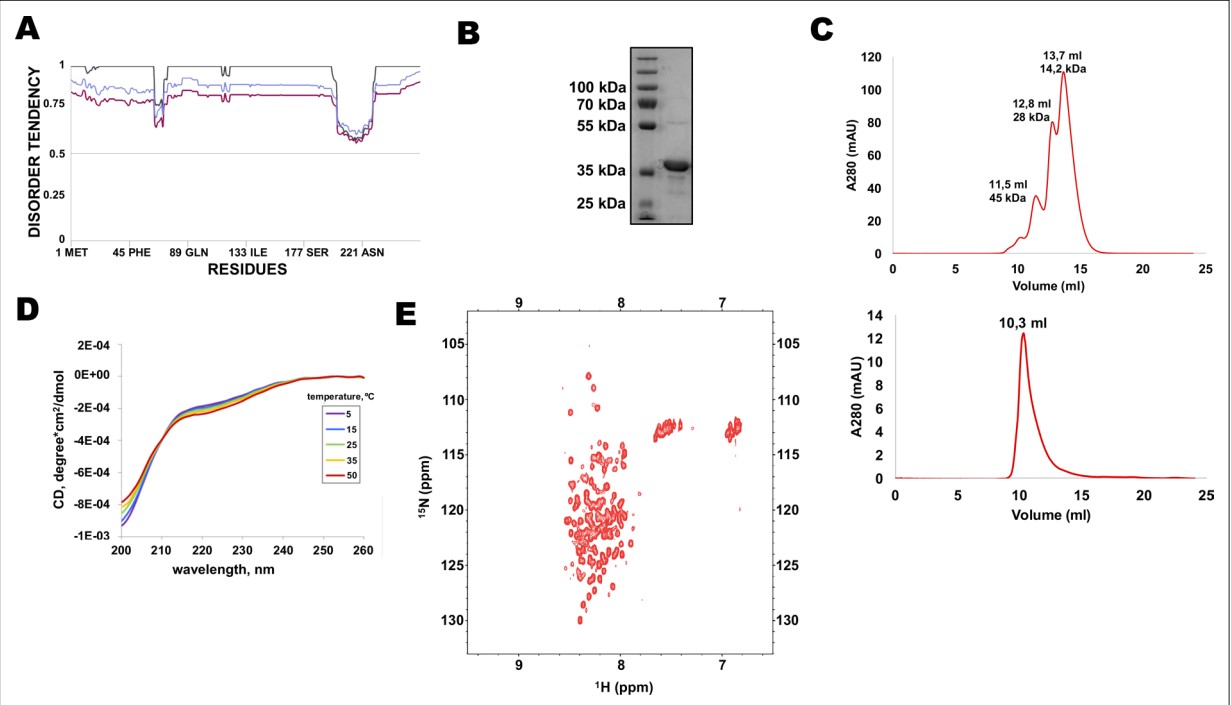

**Figure 2.** N-terminal extension of HpRif1 is intrinsically disordered. (**A**) Disorder prediction by Metadisorder (*Kozlowski and Bujnicki, 2012*) for 1–264 fragment of HpRif1 using three different combinations of algorthyms: MetaDisorder (black), MetaDisorderMD (purple), MetaDisorderMD2 (blue). All residues with the disorder probability over 0.5 are considered to be disordered. (**B**) An aliquot of the purified recombinant Rif1$_{NTE}$$^{1-264}$ was analyzed by SDS-PAGE and Coomassie staining. (**C**) Profiles of size-exclusion chromatography of the mixture of the standard proteins (upper profile) and the recombinant Rif1$_{NTE}$$^{1-264}$ (lower profile). (**D**) Circular dichroism spectra of the Rif1$_{NTE}$$^{1-264}$ recorded at 5 °C (purple), 15 °C (blue), 25 °C (green), 35 °C (yellow), and 50 °C (red). (**E**) Two-dimensional $^1$H-$^{15}$N HSQC NMR spectrum of $^{15}$N-labeled Rif1$_{NTE}$$^{1-264}$ recorded at 25 °C.

The online version of this article includes the following source data for figure 2:

**Source data 1.** The original (raw unedited) gels/blots for *Figure 2*.

---

supplement 1B). On the contrary, Rif1 association with the internal telomeric repeats is weak and close to the background levels (ChIP WSC3 signal is ~8 fold lower than TEL, *Figure 1—figure supplement 1C*). Therefore, we conclude that interaction with Rap1 is not a major determinant of telomeric localization of Rif1 in *H. polymorpha*.

## NTE aids in recruitment of HpRif1 at telomeres

In an attempt to identify portions of HpRif1 important for its telomeric localization, we substituted the *RIF1* locus with several HA-tagged C-terminal truncation constructs of Rif1. However, the only variant showing robust expression was the 1–264 fragment containing residues from the NTE region of HpRif1. Quite unexpectedly, we found that this Rif1$_{NTE}$$^{1-264}$ fragment can be efficiently recruited to telomeres, according to our ChIP experiments (*Figure 1D*). Moreover, removal of residues 1–264 from HpRif1 (Rif1$^{\Delta NTE}$) strongly reduced its ability to associate with telomeric DNA (*Figure 1D*), and led to telomere elongation (*Figure 1E*), while having no effect on the accumulation of the protein (*Figure 1F*). Thus, Rif1$_{NTE}$$^{1-264}$ is an important determinant of Rif1 telomeric localization in *H. polymorpha*.

## Rif1$_{NTE}$$^{1-264}$ is intrinsically disordered in *H. polymorpha*

Multiple protein sequence alignment of Rif1 homologues from distantly related budding yeasts revealed that N-terminal extensions have no significant similarity (*Supplementary file 1*), suggesting that Rif1 NTE may not have any conserved folding. Moreover, MetaDisorder prediction (*Kozlowski and Bujnicki, 2012*) for 1–264 fragment of HpRif1 suggests that it lacks any secondary structure (*Figure 2A*), that is HpRif1$_{NTE}$ may be unstructured or intrinsically disordered. It is worth noting that during gel-filtration and denaturing gel electrophoresis 28 kDa Rif1$_{NTE}$$^{1-264}$ behaves like a protein with higher molecular weight (*Figure 2B and C*), which is common for many intrinsically disordered

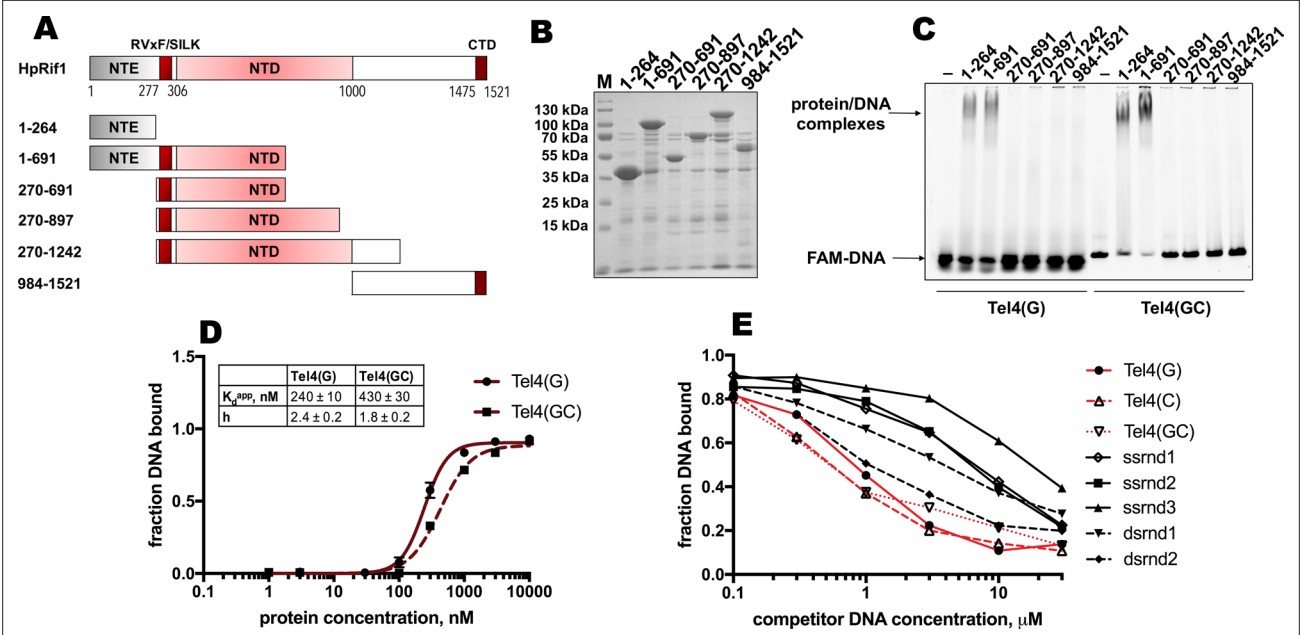

**Figure 3.** Rif1$_{NTE}^{1-264}$ binds DNA in vitro. (**A**) Schematic illustration of the 6His-S-tagged Rif1 fragments expressed in *E. coli* and used in EMSA assays. (**B**) Aliquots of the Ni-NTA purified fragments were analyzed by SDS-PAGE and Coomassie staining. "M" – protein weight marker. (**C**) 5 μM of the indicated 6His-S-tagged HpRif1 fragments were subjected to the electrophoretic mobility shift assay, using 0.5 μM of either ss- (Tel4(G)) or ds- (Tel4(GC)) DNA oligonucleotide comprising four telomeric repeats as a probe. '–' – no protein control. Positions of the free DNA and protein/DNA complexes are indicated by arrows. (**D**) Quantification of the titration EMSA experiment (two replicates) with increasing concentration of recombinant (tag-free) Rif1$_{NTE}^{1-264}$ (0, 1, 3, 10, 30, 100, 300, 1000, 3000, 10,000 nM) and 30 nM FAM-Tel4(G) (black circles) or FAM-Tel4(GC) (black squares) as probes (gels are shown in *Figure 3—figure supplement 1A*). The fits into the 'Specific binding with Hill slope' model are shown in dark red (FAM-Tel4(G) – solid curves; FAM-Tel4(GC) – dashed curves). The best-fit values for Kd apparent and Hill coefficient are shown. (**E**) Quantification of the competition EMSA experiment (the correspondent gels are shown in *Figure 3—figure supplement 1B*) with 30 nM FAM-Tel4(G), 1 μM (tag-free) Rif1$_{NTE}^{1-264}$ and increasing concentration of competitor DNA oligonucleotides (0.1, 0.3, 1, 3, 10, 30 μM). The sequences of the competitors are in *Table 1*. We note that competition with the Tel4(C) oligo may be difficult to interpret since it may first titrate out the FAM-Tel4(G) probe.

The online version of this article includes the following source data and figure supplement(s) for figure 3:

Source data 1. Numerical data used to generate *Figure 3*.

Source data 2. The original (raw unedited) gels/blots for *Figure 3*.

Figure supplement 1. EMSA experiments.

Figure supplement 1—source data 1. The original (raw unedited) gels/blots for *Figure 3—figure supplement 1*.

proteins (IDPs) (*Tompa, 2002*). To directly examine the Rif1$_{NTE}^{1-264}$ structure we applied biophysical methods. Circular dichroism spectra of Rif1$_{NTE}^{1-264}$ recorded at different temperatures demonstrated that it contains no significant α-helical or β-strand secondary structural elements over this temperature range, as judged by the absence of well-defined peaks in the 215–230 nm region (*Figure 2D*). The isoelliptical point near 210 nm is indicative of the equilibrium between left-handed PPII and truly unordered conformation, which has been often observed in CD spectra of IDPs (*Bienkiewicz et al., 2000*). We also recorded two-dimensional $^1$H-$^{15}$N HSQC NMR spectrum of $^{15}$N-labeled Rif1$_{NTE}^{1-264}$. Narrow chemical shift dispersion of amide protons (7,8–8,6 ppm) in the spectrum confirms that Rif1$_{NTE}^{1-264}$ is intrinsically disordered (*Figure 2E*).

## Two 'clusters' within Rif1$_{NTE}$ are important for DNA binding in vitro and promote telomeric localization in vivo

DNA binding by Rif1 has been demonstrated in several species (including mammals, fission, and budding yeasts; *Kanoh et al., 2015*; *Mattarocci et al., 2017*; *Moriyama et al., 2018*; *Sukackaite et al., 2014*; *Xu et al., 2010*). We decided to test whether Rif1-DNA interaction is important for its telomeric localization in *H. polymorpha*. We divided HpRif1 into several parts (*Figure 3A*) and expressed them as 6His-S-tagged recombinant proteins in *E. coli*. After affinity chromatography on

**Table 1.** Oligonucleotides used in the EMSA experiments.

| Oligo name | Sequence (5'–3') |
| --- | --- |
| Tel4(G) | GGGTGGCGGGGTGGCGGGGTGGCGGGGTGGCG |
| Tel4(C) | CGCCACCCCGCCACCCCGCCACCCCGCCACCC |
| Tel4(GC) | annealed from Tel4(G) and Tel4(C) |
| ssrnd1 | ACGACTCACTGTAGATACGACTCACTGTAGAT |
| ssrnd2 | ATCTACAGTGAGTCGTATCTACAGTGAGTCGT |
| ssrnd3 | AAATCTAGACATGAAAAAAAAAATGTTAGTAATCGAAATCTC |
| dsrnd1 | annealed from ssrnd1 and ssrnd2 |
| dsrnd2antisense | GAGATTTCGATTACTAACATTTTTTTTTTCATGTCTAGATTT |
| dsrnd2 | annealed from ssrnd3 and dsrnd2antisense |

Ni-NTA resin, we obtained protein preparations significantly enriched with the HpRif1 fragments (*Figure 3B*). We analyzed the ability of these fragments to interact with ss- and ds-DNA oligonucleotides (FAM-Tel4(G) and FAM-Tel4(GC), respectively) comprised of four telomeric repeats using EMSA. We found that only two fragments (both containing amino acids 1–264 of HpRif1) could bind telomeric DNA (both single- and double-stranded, *Figure 3C*). Thus, HpRif1 contains a DNA-binding domain at its N-terminus.

The titration experiments revealed that affinities for telomeric ssDNA and dsDNA are very close ($Kd_a^{PP}$ ~ 240 nM for FAM-Tel4(G), and $Kd_a^{PP}$ ~ 430 nM for FAM-Tel4(GC)) in our experimental conditions (*Figure 3D*, *Figure 3—figure supplement 1A*). The competition EMSA confirmed that $Rif1_{NTE}^{1-264}$ poorly differentiates between Tel4(G), Tel4(C) (C-rich telomeric strand) and Tel4(GC) oligonucleotides (*Figure 3E*, *Figure 3—figure supplement 1B*). However, it also revealed that $Rif1_{NTE}^{1-264}$ has some preference towards telomeric substrates compared with the G/C-poor DNA oligonucleotides (*Figure 3E*, *Figure 3—figure supplement 1B*, *Table 1*).

Next, we sought to identify residues within the N-terminal extension of HpRif1 which are involved in DNA binding, in order to provide evidence for direct interaction between HpRif1 and telomeric DNA in living cells. Lack of conservation and intrinsic disorder of $Rif1_{NTE}^{1-264}$ precluded us from predicting DNA-contacting residues by comparison with other known DNA-binding proteins. We tested several NTE truncation constructs for their ability to shift telomeric DNA in vitro and found amino acids 101–264 to be dispensable for DNA-binding activity (*Figure 4—figure supplement 1*). Then we noticed that region 1–100 of HpRif1 contains two similar clusters enriched in positively charged residues ($^{38}$KRNNRSR$^{44}$ and $^{79}$KRSTNNKSK$^{87}$, *Figure 4A*). Lysines and arginines often mediate DNA-protein contacts, and we presumed that these two clusters might be responsible for the observed $Rif1_{NTE}^{1-264}$-DNA interaction. We expressed and purified three mutant 6His-S-$Rif1_{NTE}^{1-264}$ proteins (*Figure 4A and B*): two with four alanine substitutions in either cluster 1 or cluster 2 ($^{38}$AANNASA$^{44}$ or 4 A$^1$, $^{79}$AASTNNASA$^{87}$ or 4 A$^2$) and one with eight alanine substitutions ($^{38}$AANNASA$^{44}$ /$^{79}$AASTNNASA$^{87}$ or 8 A). According to the results of the EMSA experiments, $8ARif1_{NTE}^{1-264}$ retained little (if any) ability to bind DNA in vitro (*Figure 4C*). Thus, we conclude that regions 38–44 and 79–87 are crucial elements of DNA-binding activity of $Rif1_{NTE}^{1-264}$ in vitro.

Then effect of the mutations in N-terminal extension of HpRif1 on its telomeric functions in vivo was tested. Even four alanine substitutions (in either motif) reduce HpRif1 association with telomere VII to the background levels (*Figure 4D*), while having little effect on the protein abundance (*Figure 4E*). Telomeres in all three mutant strains were elongated, and comparable to the $Rif1^{\Delta NTE}$ strain (*Figure 4F*). Collectively, these data demonstrate an important role of regions 38–44 and 79–87 for HpRif1 function and argue that DNA-binding activity of the N-terminal extension helps to recruit HpRif1 to telomeres.

ScRif1 binds DNA in vitro via its crook-shaped structural NTD, however we did not observe DNA binding activity of the HpRif1 fragments corresponding to this region of the protein (*Figure 3*). This may be explained by the difficulties to correctly fold large protein fragments in a heterologous system, and we tried to find whether DNA-contacting residues from ScRif1 NTD are conserved in HpRif1.

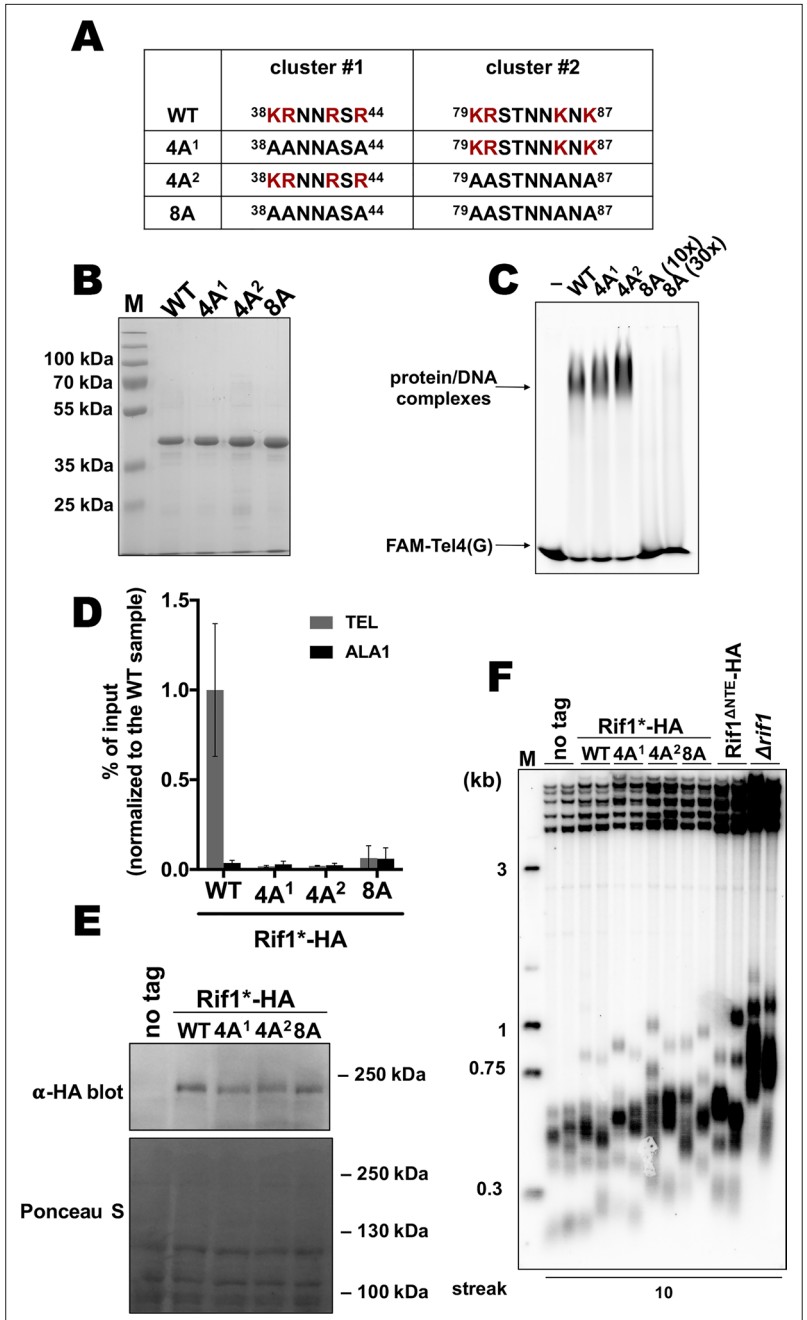

**Figure 4.** Two K/R clusters within Rif1 are important for its telomeric localization. (**A**) Sequences of the two clusters enriched in positively charged residues in the wild type and mutant versions of HpRif1. (**B**) Aliquots of the Ni-NTA purified wild type ('WT') and mutant 1–264 fragments of HpRif1 were analyzed by SDS-PAGE and Coomassie staining. 'M' – protein weight marker. (**C**) 10 μM of the indicated 6His-S-tagged proteins were subjected to EMSA, using 1 μM of ss-oligonucleotide comprising four telomeric repeats (FAM-Tel4(G)) as a probe. '–' – no protein control. '30x' – 30 μM of the 8 A mutant protein. Positions of the free DNA and protein/DNA complexes are indicated by arrows. (**D**) ChIP analysis of the indicated strains, same as in **Figure 1D**; the % of input of the 'TEL' Rif1*-HA WT sample was set to 1. Error bars indicate SD, n = 3. (**E**) Western blot analysis of the total proteins isolated from the indicated strains using antibodies targeting HA epitope (upper panel). Ponceau S-stained membrane (lower panel) served as a loading control. (**F**) Southern blot analysis as in **Figure 1B** but with different strains.

The online version of this article includes the following source data and figure supplement(s) for figure 4:

**Source data 1.** Numerical data used to generate **Figure 4**.

*Figure 4 continued on next page*

*Figure 4 continued*

**Source data 2.** The original (raw unedited) gels/blots for *Figure 4*.

**Figure supplement 1.** Rif1$_{NTE}$ truncation constructs were purified from *E.*

**Figure supplement 1—source data 1.** The original (raw unedited) gels/blots for *Figure 4—figure supplement 1*.

**Figure supplement 2.** Analysis of the HpRif1 residues corresponding to the DNA-contacting residues from ScRif1 NTD.

**Figure supplement 2—source data 1.** Numerical data used to generate *Figure 4—figure supplement 2*.

**Figure supplement 2—source data 2.** The original (raw unedited) gels/blots for *Figure 4—figure supplement 2*.

According to the published ScRif1$_{NTD}$/DNA structure, 19 charged residues have the potential to contact the DNA backbone (*Mattarocci et al., 2017*). Of these 19, only four are conserved in *H. polymorpha* Rif1 (*Figure 4—figure supplement 2A*, *Supplementary file 1*). Double mutant HpRif1$_{K658E/K666E}$ is expressed at considerably lower levels than WT HpRif1, indicating that residues K658 and K666 are important for protein stability (*Figure 4—figure supplement 2B*). Substitution of the other two conserved amino acids (K504 and R539) for glutamines does not lead to detectable changes in Rif1 telomere occupancy or telomere length (*Figure 4—figure supplement 2C,D*). Thus, HpRif1 may lack the DNA-binding mode described for ScRif1 or it utilizes different amino acids for such interaction.

## Rif1 telomere localization depends on the Ku heterodimer

To further investigate the mechanism of Rif1 recruitment, we sought to identify protein-protein interactions in which Rif1 may be involved at *H. polymorpha* telomeres. Rif1 and several other telomeric proteins were tested in yeast-two-hybrid (Y2H) assay, which revealed two Rif1-interacting partners: Ku80 and Stn1 (*Figure 5A*). Interestingly, the interaction with Ku80 (but not with Stn1) appears to be mediated through the Rif1$_{NTE}$$^{1-264}$ fragment, according to the Y2H analysis (*Figure 5A*). By testing several deletion constructs of the Rif1$_{NTE}$$^{1-264}$, we found that the residues required for Ku80 binding are localized within the 220–240 region of Rif1 (hereafter referred to as Rif1 Ku80-binding motif, or Rif1$_{KBM}$) (*Figure 5—figure supplement 1*). This region is fairly conserved in five species closely related to *H. polymorpha* (*Figure 5B*, *Supplementary file 2*), and substitution of either F225 or R230 with a negatively charged glutamic acid residue abolish the Rif1$_{NTE}$$^{150-264}$-Ku80 interaction in the Y2H system (*Figure 5C*).

Rif1-Ku interaction can also be observed in vivo by Co-IP experiments (*Figure 5D*). Deletion of either *KU80* or *KU70* led to a ~ 4 fold reduction in Rif1-HA telomere occupancy (*Figure 5E*), consistent with the idea that binding to Ku is important for Rif1 telomere association. In agreement with the Y2H data, Rif1$_{NTE}$$^{1-264}$ fragments with either F225E or R230E mutation were unable to localize to telomeres (*Figure 5F and G*), confirming that Rif1$_{NTE}$-Ku80 interaction does occur in vivo. However, F225E and R230E substitutions did not affect the ability of the full-length Rif1 to bind telomeric DNA (*Figure 5H*). Double mutation F225E/R230E and substitution of amino acids 225–230 with alanines (the 6A mutant) also did not perturb Rif1 telomere binding (*Figure 5H*). Moreover, neither mutation led to a noticeable telomere length increase (*Figure 5I*). Thus, Rif1 contains other sites for Ku binding which are more important in vivo than Rif1$_{NTE}$-Ku80 interaction.

The idea that Rif1 is recruited at telomeres by Ku relies on the assumption that Ku itself binds telomeric DNA. To verify this, we performed ChIP experiments with the Ku70-HA and Ku80-HA strains, which showed that telomeric DNA is co-immunoprecipitated with Ku (*Figure 5—figure supplement 2A*). Moreover, deletion of either Ku component perturbs telomere length maintenance (*Figure 5—figure supplement 2B*), confirming that Ku heterodimer is indeed a part of telomeric chromatin in *H. polymorpha*. Deletion of *RIF1* leads to only ~2 fold reduction in Ku80 occupancy (*Figure 5—figure supplement 2C*), in agreement with the idea that Rif1 is more reliant on the Ku's presence at telomeres than Ku on Rif1's. However, Ku loss leads to telomere shortening (~25–50% reduction in telomere length, *Figure 5—figure supplement 2B*, *Figure 5J*), indicating to a positive role of Ku in telomere lengthening. To verify that Rif1 telomere localization defect upon Ku loss is not simply a consequence of the telomere shortening, we measured Rif1 telomere binding in a strain lacking telomerase RNA (*Figure 5—figure supplement 2D*, E). TER knock-out resulted in ~40% reduction in telomere length, whereas Rif1-HA telomere occupancy diminished only ~2 fold (*Figure 5—figure supplement 2D*, E); contrasting to ~4 fold Rif1 ChIP signal drop in case of Ku mutants (*Figure 5E*).

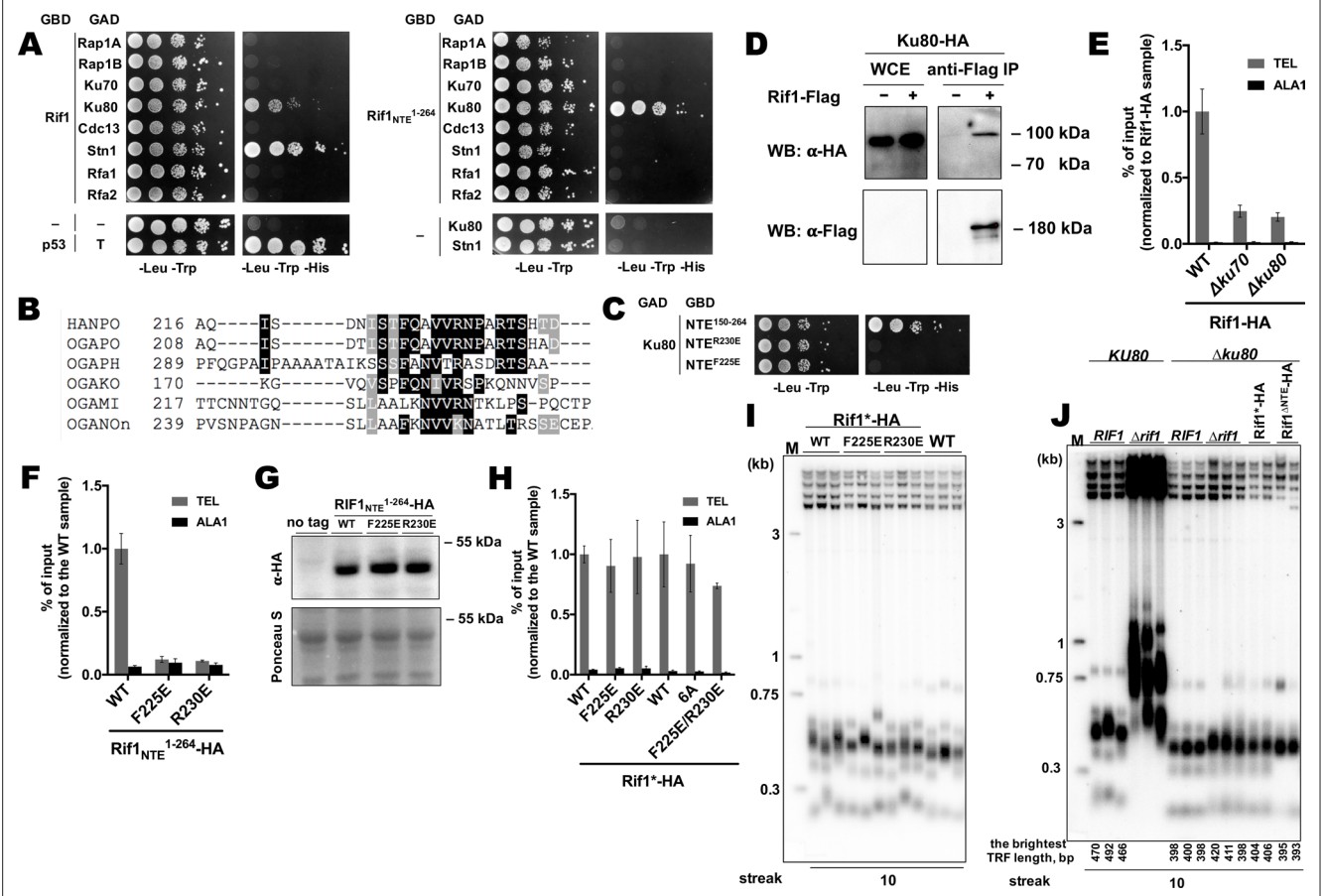

**Figure 5.** Rif1 interacts with Ku80 in *H.polymoprha*. (**A**) Y2H analysis. AH109 colonies expressing pairs of the indicated proteins (fused to either Gal4-BD (GBD) or Gal4-AD (GAD)); cultures with A₆₀₀ ~0.5 and four 10-fold serial dilutions were plated on the SC medium lacking amino acids as indicated, and incubated at 30°C for 4 days. "T" – SV40 large T antigen. (**B**) A fragment of the alignment of the NTE regions from *H. polymorpha* DL-1 (HANPO) and five of its closest relatives. Full alignment is in *Supplementary file 2*. (**C**) Y2H analysis as in (**A**), 'NTE$^{R230}$' and 'NTE$^{F225E}$' – 150–264 fragments of Rif1 with the R230E and F225E mutations, respectively. (**D**) Co-IP analysis. IP on the anti-Flag resin. The amount of tagged proteins in whole cell extracts (WCE) and the IP samples (IP) was monitored by Western blot (WB) using anti-Flag and anti-HA antibodies. The IP experiment was performed in the presence of benzonase nuclease. (**E, F**) ChIP analysis of the indicated strains, same as in *Figure 1D*; the % of input of the 'TEL' Rif1-HA WT sample was set to 1 (**E**); the % of input of the 'TEL' Rif1$_{NTE}$$^{1-264}$-HA WT sample was set to 1 (**F**). (**G**) Western blot analysis. Same as in *Figure 1F*, but with different strains. (**H**) ChIP analysis of the indicated strains, same as in *Figure 1D*; Error bars indicate SD, n = 3. '6 A' mutation: $^{225}$FQAVVR$^{230}$/$^{225}$AAAAAA$^{230}$. (**I, J**) Southern blot analysis as in *Figure 1B* (**J**) Mean lengths of the brightest TFR bands are: *RIF1KU80* 476 bp, *RIF1Δku80* 399 bp. WT telomere length reported to be ~160 bp (~20 telomeric repeats, *Sohn et al., 1999*), therefore telomere length is reduced by ~50% in the knockout strain.

The online version of this article includes the following source data and figure supplement(s) for figure 5:

**Source data 1.** Numerical data used to generate *Figure 5*.

**Source data 2.** The original (raw unedited) gels/blots for *Figure 5*.

**Source data 3.** The original (raw unedited) gels/blots for *Figure 5*.

**Figure supplement 1.** Y2H analysis.

**Figure supplement 2.** Additional experiments on the Rif1-Ku interaction.

**Figure supplement 2—source data 1.** Numerical data used to generate *Figure 5—figure supplement 2*.

**Figure supplement 2—source data 2.** The original (raw unedited) gels/blots for *Figure 5—figure supplement 2*.

**Figure supplement 2—source data 3.** The original (raw unedited) gels/blots for *Figure 5—figure supplement 2*.

Interestingly, *RIF1* deletion in a *Δku80* background does not lead to strong telomere elongation compared to the parental *Δku80* strain, suggesting that inhibition of telomerase by Rif1 is attenuated in the absence of Ku (*Figure 5J*). Moreover, inhibition by the Rif1$_{NTE}$$^{1-264}$ fragment is lost completely: no telomere lengthening is observed in the Rif1$^{ΔNTE}$*Δku80* strain compared to the *Δku80* strain

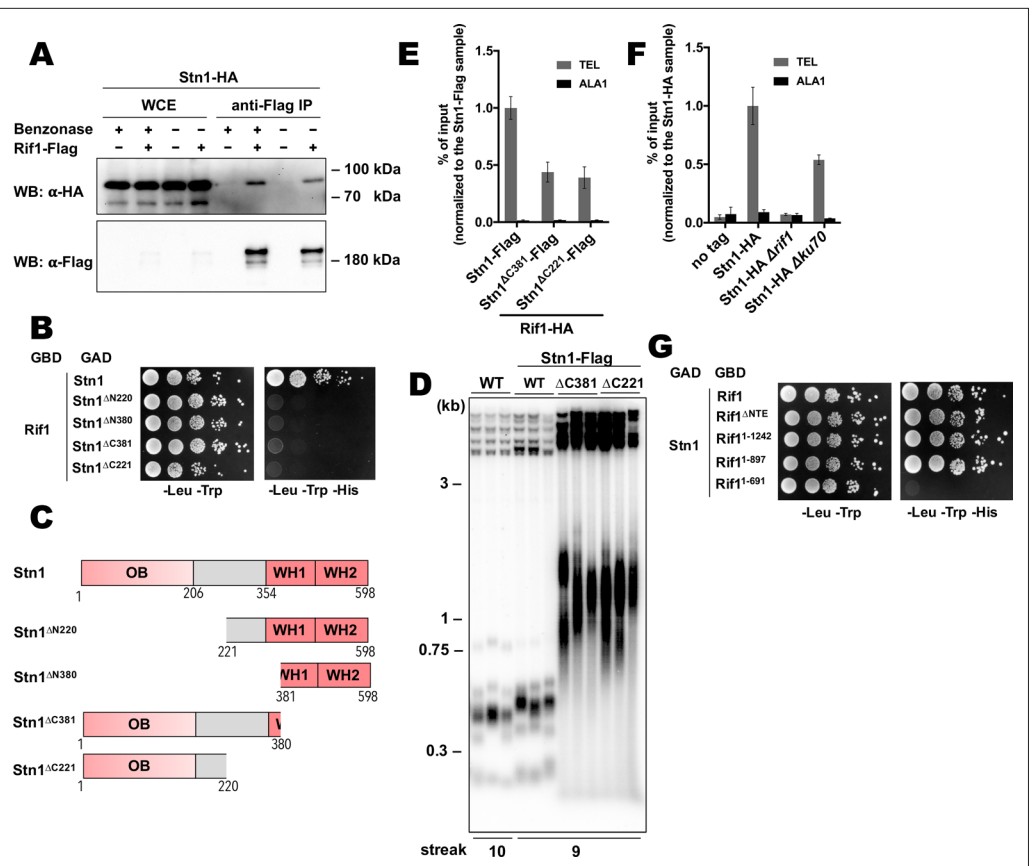

**Figure 6.** Rif1 recruits Stn1 at *H.polymorpha* telomeres. (**A**) Co-IP analysis. Same as in *Figure 5I*, but with different strains; the IP experiment was performed either with or without benzonase treatment as indicated. (**B**) Y2H analysis. Same as in in *Figure 5A*, but with different protein pairs. (**C**) Schematic illustration of the domain organization of the full-length HpStn1 and its truncation variants used in this study. (**D**) Southern blot analysis as in *Figure 1B* but with different strains. (**E, F**) ChIP analysis of the indicated strains, same as in *Figure 1D* (IP on the anti-HA beads); the % of input of the 'TEL' Rif1-HA/Stn1-Flag sample was set to 1. Error bars indicate SD, n = 3. (**E**) Or the % of input of the 'TEL' Stn1-HA sample was set to 1 (**F**). (**G**) Y2H analysis. Same as in in *Figure 5A*, but with different protein pairs.

The online version of this article includes the following source data and figure supplement(s) for figure 6:

**Source data 1.** Numerical data used to generate *Figure 6*.

**Figure supplement 1.** Cdc13, Stn1 and Ten1 are essential for viability in *H. polymorpha*.

**Figure supplement 2.** Schematic representation of the putative protein-protein interactions at *H.polymorpha* telomeres.

(*Figure 5J*). These results suggest that Ku potentially recruits some telomere addition factor, while the Rif1 fragment may compete with this putative factor for Ku80 binding. In *S. cerevisiae*, Ku is one of the accessory subunits of telomerase stably bound to its RNA component. We found that HpTER cannot be efficiently co-purified with Ku80-HA, while it robustly binds TERT-HA, Est1-HA, and Est3-HA in the same experimental conditions (*Figure 5—figure supplement 2F*, *Shepelev et al., 2020*), suggesting that HpKu is not a stable telomerase component. This does not exclude, however, the possibility of weak or transient association between Ku and telomerase in *H. polymorpha*. Consistent with this idea, we observe weak but noticeable interaction between HpTERT and HpKu80 in the Y2H assay (*Figure 5—figure supplement 2G*).

## Rif1 is important for recruitment of Stn1 protein to telomeres

Next, we decided to study telomeric role of the interaction between Rif1 and Stn1. First, we confirmed the ability of the proteins to form a complex in vivo by the Co-IP experiment. Stn1-HA co-elutes

with Rif1-HA following immunoprecipitation on the anti-Flag resin (*Figure 6A*). Benzonase treatment of the extracts prior to the IP does not diminish the amount of co-precipitated Stn1 (*Figure 6A*), suggesting that the Rif1-Stn1 interaction is DNA-independent. We were unable to test the effect of the *STN1* knockout on the Rif1 telomere binding, since Stn1 is essential for cell viability (as two other components of the CST complex – Cdc13 and Ten1, *Figure 6—figure supplement 1A*). However, we found that removal of either N- or C-terminal Stn1 domains is sufficient to completely disrupt binding to Rif1 in the Y2H assay (*Figure 6B and C*). C-terminal truncations in Stn1 are well tolerated by cells, although the Stn1$^{\Delta C381}$-Flag and Stn1$^{\Delta C221}$-Flag strains have greatly elongated telomeres (*Figure 6D*), similar to the Stn1 truncation mutants in *S. cerevisiae* (*Petreaca et al., 2007*; *Petreaca et al., 2006*; *Puglisi et al., 2008*). Rif1 telomere association is reduced in the strains with these *stn1* mutations, although the reduction is relatively modest (~2.5 fold, *Figure 6E*). Interestingly, we found that Stn1 telomere binding is completely abolished upon *RIF1* deletion, suggesting that it is actually Rif1 that recruits Stn1 to telomeres, and not vice versa (*Figure 6F*). Stn1 telomere occupancy is also diminished (~2 fold) in the *Δku80* strain, which has smaller amount of the telomere-bound Rif1 (*Figure 6F*). Finally, we found that Rif1 bind Stn1 in Y2H through its conserved NTD, more precisely – the 265–897 region (*Figure 6G*).

## Discussion

We found that Rif1 from the methylotrophic yeast *H. polymorpha* prevents hyperelongation of telomeres, similar to its other yeast counterparts. In *S. cerevisiae*, Rif1's function strongly depends on the recruitment by Rap1 protein (specifically, by the Rap1$^{RCT}$ domain). Hence, telomerase-dependent telomere overelongation can be observed in the strains with either deletion of the *RIF1* gene, or mutations that disrupt Rap1-telomere association (*Hardy et al., 1992*; *Shi et al., 2013*; *Teng et al., 2000*). Long telomeres in the *Δrif1* strain of *H. polymorpha* do appear to be solely maintained by telomerase (*Figure 1B and C*). *H. polymorpha* has two Rap1 paralogues, however, HpRap1A does not recognize telomeric DNA and RCT removal from Rap1A has no effect on telomere length (*Malyavko et al., 2019*). HpRap1B associates with double-stranded telomeric DNA and the 50 a.a. C-terminal truncation (Rap1B$^{1-526}$) leads to more than 10-fold drop in Rap1B expression and strong telomere elongation. In contrast to the *Δrif1* strain, longer telomeres in this B$^{1-526}$ strain were found to depend primarily on recombination (*Malyavko et al., 2019*). This phenotypic discrepancy indicates that functions of Rap1 and Rif1 in *H. polymorpha* may not be as intimately linked as in *S. cerevisiae*. In addition, we could not identify the Rap1-binding motif within HpRif1. We failed to detect a direct interaction between HpRif1 and either of the two HpRap1 paralogues in the Y2H experiment (*Figure 5A*). Finally, Rap1B can efficiently localize to internal as well as terminal telomeric dsDNA (as expected from a sequence specific telomeric factor), whereas Rif1 has a clear preference for telomeric repeats located at the end of the chromosome (*Figure 1—figure supplement 1*). Thus, although we cannot exclude the possibility of an interaction between HpRap1 and HpRif1, we believe that it is highly unlikely that Rif1 binding to Rap1 is the major mechanism of Rif1 telomeric recruitment in *H. polymorpha* in a manner it is described for its *S. cerevisiae* homologues.

Rif1 homologues from many species contain a DBD, and DNA binding by the NTD of *S. cerevisiae* Rif1 promotes its accumulation at telomeres in a Rap1-independent fashion (*Kanoh et al., 2015*; *Mattarocci et al., 2017*; *Moriyama et al., 2018*; *Sukackaite et al., 2014*; *Xu et al., 2010*). This prompted us to test whether HpRif1 might be recruited at telomeres by means of direct interaction with telomeric DNA. We found that HpRif1 fragments containing amino acids 1–264 possess DNA-binding activity in vitro. The Rif1$_{NTE}$$^{1-264}$ fragment can be detected at telomeres by ChIP, while its removal (the Rif1$^{\Delta NTE}$ strain) strongly diminishes the telomere binding of Rif1. Moreover, mutations that abolish DNA-binding in vitro lead to the substantial (if not complete) loss of HpRif1 from telomeres and telomere elongation comparable to the Rif1$^{\Delta NTE}$ strain (*Figure 4*). These results are in favor of the 'recruitment via direct telomeric DNA binding' hypothesis. The identified DNA-binding mode by the intrinsically disordered portion of HpRif1, however, is clearly different from the one utilized by the conserved structured NTD of ScRif1. We could not detect a DNA binding activity of the HpRif1$_{NTD}$ (residues ~ 310–1000, *Figure 3A–C*), although this may be due to improper folding of the large polypeptides in the *E. coli* expression system or suboptimal conditions of the EMSA reactions. Despite the fact that the NTD is a relatively well conserved region of Rif1, we found that of 19 charged amino acids constituting the DBD of ScRif1 only four can be found in HpRif1 and their mutation has no apparent

effect on Rif1-telomere association (*Figure 4—figure supplement 2*). Given the high evolvability of telomeric proteins, we do not, however, dismiss the possibility that HpRif1 may contact DNA through different amino acids within its NTD domain.

Interestingly, there is a discrepancy between our ChIP experiments, which suggest that Rif1$_{NTE}$ is necessary and sufficient for the telomeric localization of Rif1, and the telomere length analysis, which points to the relatively minor contribution of the Rif1$_{NTE}$ to the telomere length control (*Figure 4D and F*). This points to the existence of another mode(s) of HpRif1 recruitment, which may evade detection by ChIP, perhaps, because it is restricted to the short window of the cell cycle or due to its transient nature. It is important to note, that the ChIP method proved not to be sufficiently sensitive in assessment of Rif1 chromatin binding in *S. cerevisiae*: Rif1 accumulation at replication origins, for example, was revealed only by the chromatin endogenous cleavage (ChEC) method (*Hafner et al., 2019*; *Hafner et al., 2018*). Future experiments utilizing more sensitive techniques may reveal other modes of HpRif1 recruitment, which could be attributed to the DNA binding by the NTD (similar to the ScRif1) and/or to the interaction of HpRif1 with other telomeric proteins.

Our search for Rif1's protein partners at telomeres revealed two hits – Ku80 and Stn1 – and deletion of the *KU80* gene does lead to strong reduction in Rif1 telomere occupancy, suggesting that this interaction plays an important role in localizing Rif1 to telomeres (*Figure 5*). Although in the Y2H assay the Ku80-Rif1 interaction depends primarily on the short motif within Rif1$_{NTE}$, our experiments indicate that in *H. polymorpha* cells the robust Ku80-Rif1 association requires other Rif1 domains. The Rif1$_{KBM}$ may thus serve only as an auxiliary module, which is consistent with the fact that it is conserved only in five species closest to *H. polymorpha*. The interaction of NTE-less Rif1 with Ku could therefore represent the second mode of telomeric recruitment of HpRif1 responsible for the telomerase inhibition in the Rif1$^{\Delta NTE}$ strain.

In *S. cerevisiae*, telomere tract length determines the probability of the elongation of a chromosomal end: shorter telomeres are preferred substrates for telomerase (*Marcand et al., 1999*; *Teixeira et al., 2004*). At the basis of this regulation is 'counting' of telomeric repeats by Rap1 that defines the number of (telomerase-inhibitory) Rif1 and Rif2 molecules present at any given telomere (*Levy and Blackburn, 2004*; *Marcand et al., 1997*). *H. polymorpha* lacks Rif2 entirely, and at this point it is not obvious how mechanistically Rif1 recruitment could be linked to telomere length. In fact, HpRif1 appears to have only limited ability to bind double-stranded telomeric DNA in vivo (*Figure 1—figure supplement 1*). It should be noted, that Rif1 is a large protein with many functions and, even in *S. cerevisiae*, it is not fully involved in the 'counting' process. The Rif1$^{RBM}$ mutant lacking the ability to contact Rap1 is almost completely lost from telomeres as judged by ChIP, yet retains a significant portion of its telomerase inhibitory potential (*Mattarocci et al., 2017*; *Shi et al., 2013*). Thus, budding yeast Rif1 homologues may operate (at least partially) in a telomere length-independent way.

Being the only known telomeric dsDNA binding factor in *H. polymorpha*, Rap1B is the obvious candidate for a telomere length sensor within a potential 'protein-counting model'. However, as described previously (*Malyavko et al., 2019*), Rap1B's role in telomerase inhibition (if exists) appears to be minor, and through which protein partners (and whether at all) Rap1B could be involved in inhibition of telomerase action is yet to be determined. The lack of Rif2, our inability to detect a Rap1-Rif1 interaction and the apparent absence of a prominent inhibitory effect of Rap1B on telomerase suggest that 'protein counting' mechanism of telomere length regulation described for *S. cerevisiae* may not be operating in *H. polymorpha*. Additional studies are needed to establish whether this is indeed the case and which components of the telomeric chromatin are 'counted' to provide a negative feedback loop for telomerase.

Similarly to *S. cerevisiae* (*Boulton and Jackson, 1996*; *Porter et al., 1996*), deletion of the subunits of the HpKu heterodimer leads to pronounced telomere shortening (*Figure 5—figure supplement 2B*). ScKu recognizes a specific stem-loop structure within telomerase RNA, mediating one of the two pathways of telomerase recruitment (*Chen et al., 2018*; *Peterson et al., 2001*; *Stellwagen et al., 2003*). Our previous analysis of the *H. polymorpha* telomerase RNA structure did not reveal a similar Ku-binding hairpin (*Smekalova et al., 2013*) and the lack of a stable interaction between Ku and HpTER was confirmed experimentally in this study (*Figure 5—figure supplement 2F*). Shorter telomeres in the HpKu knock-out strains could be explained, however, by a transient association with telomerase via another telomerase subunit; for example like in human cells, where an interaction between Ku and TERT has been detected (*Chai et al., 2002*). Indeed, we found that HpTERT binds HpKu80

in the Y2H experiment (*Figure 5—figure supplement 2G*). Interestingly, we observed only minor telomere elongation upon HpRif1 loss in the *Δku80* background, suggesting that Rif1 counteracts the positive effect of the Ku at telomeres (*Figure 5J*). We propose that Rif1's inhibitory function in *H. polymorpha* may involve a direct competition with telomerase for Ku binding at telomeres (*Figure 6—figure supplement 2*).

The second pathway of telomerase recruitment in *S. cerevisiae* relies on the binding of the Est1 telomerase subunit to the RD (recruitment domain) of the Cdc13 protein (*Chen et al., 2018*; *Evans and Lundblad, 1999*; *Pennock et al., 2001*). Association with Stn1 blocks this interaction, thereby limiting the amount of telomerase at telomeres (*Chandra et al., 2001*; *Li et al., 2009*; *Pennock et al., 2001*). Although HpCdc13 is shorter than ScCdc13 and lacks the RD domain (and, apparently, the ability to bind Est1), we have demonstrated recently, that it can bind both telomerase (via the TERT subunit) and Stn1 in a Y2H assay, indicating that a similar mechanism of telomerase regulation may be operating in *H. polymorpha* (*Malyavko and Dontsova, 2020*). Alternatively, Stn1 may influence telomerase binding indirectly, by controlling the amount of ssDNA (and therefore Cdc13, which is a ssDNA binding protein) at telomeres (*Grandin et al., 1997*). In any case, we found that Stn1 accumulation at telomeres depends strongly on Rif1, and that mutations disrupting Stn1-Rif1 (and Stn1-Cdc13) interaction lead to highly elongated telomeres (*Figure 6*). Thus, we speculate that the negative effect of Rif1 on telomere length may also be explained by its ability to promote Stn1 telomere localization, thereby attenuating the telomerase recruitment via Cdc13 (*Figure 6—figure supplement 2*).

Remarkably, the existence of a physical interaction between Rif1 and the CST complex was proposed (but not demonstrated experimentally) in a study by *Anbalagan et al., 2011* to explain the specific requirement of Rif1 for viability of *S. cerevisiae* cells with hypomorphic mutations in Cdc13 and Stn1. Our Y2H experiments show that Stn1 binds the NTD of Rif1 (*Figure 6G*). Rif1$_{NTD}$ contains the most conserved portion of the protein (*Supplementary file 1*) that is present in all Rif1 orthologues (*Mattarocci et al., 2017*; *Sreesankar et al., 2012*). Hence it is plausible that the Stn1-Rif1 direct interaction observed in our study may be present in other species.

In summary, in the present study we have identified several novel aspects of Rif1 function in budding yeasts. We demonstrated that the N-terminal extension (NTE) of Rif1 is an accessory domain partially promoting Rif1 telomeric localization in *H. polymorpha*. Given that DNA binding by Rif1$_{NTE}$ is weak and lacks clearly defined specificity toward a particular telomeric substrate, we believe that it may simply stabilize Rif1 at telomeres, while the preference for the telomeric chromatin is provided by the ability to contact Ku heterodimer. Consistently, Rif1$_{NTE}^{1-264(F225E)}$ and Rif1$_{NTE}^{1-264(R230E)}$ mutants (unable to bind Ku80, but with intact DBD) fail to localize to telomeres (*Figure 5G*). Our results also indirectly point to the possibility of the DNA recognition by the HpRif1$_{NTD}$ in a manner akin to ScRif1, which recognizes the 3'-overhang containing ss-ds junctions (*Mattarocci et al., 2017*). This may also be a factor driving HpRif1 to the chromosomal ends. Finally, we uncovered an HpRif1's role in loading Stn1 protein onto telomeres. Curiously, the mammalian Rif1 homologue is important for the recruitment of CST complex to promote NHEJ at several types of DSBs (although the Rif1-CST interaction is not direct, but mediated via the Shieldin complex; *Barazas et al., 2018*; *Chapman et al., 2013*; *Dev et al., 2018*; *Di Virgilio et al., 2013*; *Escribano-Díaz et al., 2013*; *Gupta et al., 2018*; *Mirman et al., 2018*; *Noordermeer et al., 2018*; *Tomida et al., 2018*; *Zimmermann et al., 2013*). It is tempting to speculate that Rif1's role in recruiting Stn1 may be conserved from yeasts to human.

# Materials and methods

## Yeast strains and constructs

The strains used in this study are listed in *Supplementary file 1*. The DL1-L strain (*Sohn et al., 1996*) was used as a wild type (no tag) control in all experiments. Gene replacements (detailed below) and random spore analysis was performed as described (*Malyavko et al., 2019*). The gene replacements were verified by PCR, mutagenesis was verified by sequencing.

One to 1001 bp region of the *RAD52* gene, 190–480 bp region of the *HpTER* gene and 196–2076 bp region of the *KU80* gene were replaced with pKAM555 plasmid (linearized with SmaI) (*Agaphonov et al., 2010*). For the *HpRIF1* knockout, –33–4430 bp region of the *RIF1* gene was replaced with *HpLEU2* gene from pCHLX vector (*Sohn et al., 1996*). For the *HpKU70* knockout, 214–1698 bp region

of the *KU70* gene was replaced with either *HpLEU2* gene from pCHLX, or pKAM555 (linearized with SmaI).

For C-terminal 3HA tagging, stop codon (or a relevant portion of the gene, in case of a concomitant C-terminal truncation) was replaced with SmaI-PmeI fragment of the pFA6a-3HA-HpURA3 plasmid (*Malyavko et al., 2019*). The pFA6a-Rif1-3HA-HpLEU2 vector was constructed by replacing the BglII/PmeI fragment of the pFA6a-Rif1-3HA-HpURA3 with the *HpLEU2* marker gene (generated by PCR from the pCHLX vector with primers F: 5'-aaaagatctccaccgcggtggc-3' and R: 5'-cggggggatcctactttttttttctcc-3'); then the 3HA sequence was replaced with 3Flag by PCR and self-ligation of the pFA6a-Rif1-3HA-HpLEU2 (primers F: 5'-caccgtcatggtctttgtagtctccacccccgcctcccccgcgtcttttttcaaacacgtc-3' and R: 5'- attataaagatcatgacatcgactacaaggatgacgatgacaagtagggcgcgccacttctaaataag-3'), generating the pFA6a-Rif1-3Flag-HpLEU2 vector. For C-terminal 3Flag tagging, stop codon (or a relevant portion of the gene, in case of a concomitant C-terminal truncation) was replaced with the 3FlagLEU2 fragment (primers F: 5'-gggggaggcggggggtggagactac-3' and R: 5'-cggggggatcctactttttttttctcc-3') of the pFA6a-Rif1-3Flag-HpLEU2.

For mutagenesis of the NTE of HpRif1 we generated the p5rCHLXpr plasmid by cloning upstream *RIF1* flanking region (primers F: 5'-aaccgcggaccaggtcatctacagagacgag-3'; R: 5'-aaatctagacgggtgtgtgattctgcaaacc-3') and *RIF1* promoter (primers F: 5'-aaggatccaaaaaccaaaaaaaatgccagctt-gaaaaaaattg-3'; R: 5'-aagaattcggcttctggttggaaaatacag-3') at SacII/XbaI sites and BamHI/EcoRI sites, respectively, of the pCHLX vector. Then *RIF1* fragments (F: 5'-aaagatatcatgagtgctaatgacaacga-cacg-3'; R: 5'-aaagtcgacaggcggactcactttcaagattg-3') and (F: 5'-aaagatatcatgcgggacgcggccggcaac-3' R: 5'-aaagtcgactccacggcgtgcaagctca-3') were cloned at SalI/EcoRV sites to generate templates for Rif1*-HA and Rif1$^{\Delta NTE}$-HA cassettes, respectively. Templates for integration cassettes with mutations in NTE were introduced by PCR amplification of the plasmid with the template for Rif1*-HA cassette and subsequent self-ligation. All resulting cassettes were PCR amplified and transformed into Rif1-HA strain to generate Rif1*-HA, Rif1$^{\Delta NTE}$-HA and the mutant Rif1*-HA strains.

In the heterozygous *H. polymorpha* CBS4732 strains the portions of the genes (−17–951 bp of *CDC13*; 22–1618 bp of *STN1*; 1–365 bp of *TEN1*; 78–1582 bp of *KU70*) were replaced with *HpLEU2* gene from pCHLX.

For the intTEL18 strain creation we generated the pUC19LGPH36 plasmid by cloning *leu2* gene (from DL1-L genome) with flanking regions (primers F: 5'-gcgcgtgtctcagcatgaac-3'; R: 5'-ggtgtgggaggtagaagagg-3') at the pUC19 SmaI site. The *leu2* ORF was replaced (primers F: 5'-aaaaagatctaattatactgttgcgcgaagtagtcccatggtaggatctcgaataattcctaaataatcc-3'; R: 5'-gattgcaaaat-gatggaactattttgc-3') by the PstI(blunted)-BclI fragment of the HARS36 sequence (*Sohn et al., 1999*) from the AMIpSL1 vector (*Agaphonov et al., 1999*). Then the G418$^R$ gene (primers F: 5'-aaaaaag-acaggaatgagtaaatgaagatcctttgatcttttctacgg-3'; R: 5'-ccgggaaaaactgaaaaaccattggcacgacaggtttc-ccgac-3' from the pKAM555 vector) was inserted at the ClaI site using HiFi assembly (NEB). For the intTEL0 strain creation the pUC19LG plasmid (lacking the PstI(blunted)-BclI fragment of the HARS36 sequence) was used.

## Yeast growth conditions

Normally, single colonies from the transformation plates were restreaked on YPD plates and grown for 2 days at 37°C. After 10 restreaks (~230 cell divisions) cells were grown in 10 ml of YPD at 37 °C overnight and used for Southern blot analysis.

For the Δ*ter* strains viability assay (*Figure 1B and C*) several colonies from the transformation plate were resuspended in 100 μl of water: 50 μl were used for yeast colony PCR, 2 μl were plated onto a fresh YPD plate and grown for 2 days at 37 °C ('streak 1' plate), 48 μl were added to 10 ml of YPD and grown overnight at 37 °C (used for gDNA isolation and Southern blot, 'streak 0'). Single colonies from the 'streak 1' plate were resuspended in 10 μl of water, 3 μl were plated onto a fresh YPD plate and grown for 2 days at 37 °C ('streak 2' plate), 7 μl were added to 10 ml of YPD and grown overnight at 37 °C (used for gDNA isolation and Southern blot, 'streak 1'). Single colonies from the 'streak 2' (smears of colonies in case of the Δ*ter* strains) plate were restreaked onto a fresh YPD plate and grown for 2 days at 37 °C ('streak 3' plate). The *TER* strains were restreaked seven additional times, single colonies from the 'streak 10' were grown in 10 ml of YPD overnight at 37 °C and used for Southern blot. Two colonies of each genotype were analyzed in this experiment (biological replicates).

## Telomere Southern blots

Southern blot experiments were carried out as previously described (*Shepelev et al., 2020*). TRF length were calculated using the ImageQuant TL 1D software version 7.0.

## Chromatin immunoprecipitation (ChIP)

ChIP experiments were performed as previously described (*Malyavko et al., 2019*) with the following modifications. Cells were grown in 100 ml of YPD at 37°C and fixed with 1% formaldehyde for 30 min at 25°C. Lysates were incubated with anti-HA magnetic beads for 2 hr. High-salt lysis buffer contained 0.5 M NaCl, and wash buffer did not contain any SDS. We used the standard deviation between at least three replicates (cultures from three colonies of the same strain) as a value of the experimental error.

## Co-immunoprecipitation (Co-IP) and western blots

Co-IP experiments were performed as previously described (*Shyian et al., 2020*) with the following modifications. $OD_{600}$ ~ 0.9 cell cultures were used to prepare 1 ml of lysate. Cells were broken with glass beads in a Precellys Evolution homogenizer. 20 µl of anti-Flag M2 gel (Sigma) per 1 ml of lysate were used. Proteins were eluted by boiling at 95 °C for 10 min in 30 µl of 2 x SDS-PAGE buffer (100 mM Tris pH 8, 4% SDS, 10% glycerol, 0.2% bromophenol blue). Total proteins were isolated as previously described (*Kushnirov, 2000*). Proteins were separated on 8% PAGE gels, transferred onto Hybond P 0.22 PVDF membrane (GE Healthcare), stained with Ponceau S (Amresco), and blocked in 5% BSA. Anti-HA-HRP antibodies (clone 3F10, Sigma) at 1:5000, anti-FLAG M2 antibodies (Sigma) at 1:5000, goat anti-mouse IgG-HRP (62–6520, Thermo Fisher Scientific) at 1:5000 and SuperSignal West Femto Maximum Sensitivity Substrate (Thermo Fisher Scientific) were used for protein detection.

RNA Co-IP experiments were performed as described in *Shepelev et al., 2020*.

## Protein expression and purification

Rif1 fragments for EMSA were expressed and purified as HpRap1B protein (*Malyavko et al., 2019*).

For CD and NMR studies 1–264 HpRif1 fragment was purified in 50 mM phosphate buffer, pH 7.5, 500 mM NaCl, 10 mM β-ME, 10% glycerol, 0,05% Tween 20, 30 mM Imidazole, 0,3 mM PMSF followed by ion-exchange chromatography on Heparine agarose (Sigma). Then 6His- and S-tags were excised by overnight incubation with recombinant TEV protease at 4°CC; tags and TEV protease were removed by an additional round of affinity chromatography on Ni-NTA-agarose (Sigma). As a final step gel filtration on a Superdex 75 column in 20 mM phosphate buffer, pH 7.5, 50 mM NaCl, 5% glycerol was applied. 750 µg of ovalbumin, 400 µg of carbonic anhydrase and 400 µg of lactalbumin were injected separately to serve as standards. For the $^{15}N$ isotope, labeling cells were cultivated at 37 °C in M9 minimal medium with 1 g/L $^{15}NH_4Cl$ (Cambridge Isotope Laboratories, Inc).

## Biophysical measurements

CD spectra were recorded at the following temperatures: 5, 15, 25, 35°C and 50 °C. A protein concentration of 0.3 mg/ml was used in 20 mM phosphate buffer, pH 7.5, 50 mM NaCl, 5% glycerol. CD measurements were made on a Chirascan CD spectrometer (Applied Photophysics) using a 0.1 mm path length.

The NMR samples with concentration of 0.2 mM $^{15}N$-labeled protein were prepared in 90% $H_2O$/10% $D_2O$, 50 mM NaCl, and 20 mM sodium phosphate buffer (pH 7.2). Spectra were acquired at 298 K on a Bruker Avance 600 MHz spectrometer equipped with a triple resonance ($^1H$, $^{13}C$ and $^{15}N$) pulsed field z gradient probe. NMR spectra were processed and analyzed using the Mnova software (Mestrelab Research, Spain).

## Electrophoretic mobility shift assay (EMSA)

The sequences of the oligonucleotides used in EMSA are in *Table 1*. EMSA experiments were performed as described (*Malyavko et al., 2019*) with the following modification. Reaction buffer contained 10 mM HEPES-NaOH, pH 7.5, 100 mM NaCl, 0.5 mM DTT, 0.25 mg/ml bovine serum albumin, 5% glycerol. Band intensities (BI) were quantified in ImageQuant TL 7.0, values of 'fraction DNA bound' (FDB) were calculated using formula FDB = BI*(complex)/(BI*(complex)+ BI(free DNA)),

where BI*(complex) is BI of the complex minus BI of the corresponding area in the 'no protein control' lane. The fits into the 'Specific binding with Hill slope' were done using GraphPad Prism version 7.00.

## Yeast two-hybrid (Y2H) system

Y2H experiments were carried out as previously described (*Malyavko and Dontsova, 2020*).

## Acknowledgements

We thank M Agaphonov (Institute of Experimental Cardiology, Cardiology Research Centre, Moscow, Russia) who kindly provided strains, plasmids and protocols for *H. polymorpha* genetics, A Arutyunyan (Belozersky Institute of Physico-Chemical Biology, Lomonosov Moscow State University) for CD spectra measurements. The study was supported by the Russian Foundation for Basic Research [13-04-40199-Н, 17-04-01692-A] and Interdisciplinary Scientific and Educational School of Moscow University «Molecular Technologies of the Living Systems and Synthetic Biology». Biochemical studies were supported by the Russian Science Foundation [21-64-00006]. NMR studies were supported by the Russian Science Foundation [19-14-00115].

## Additional information

### Funding

| Funder | Grant reference number | Author |
| --- | --- | --- |
| Russian Foundation for Basic Research | 13-04-40199-Н | Olga A Dontsova |
| Russian Foundation for Basic Research | 17-04-01692-A | Alexander N Malyavko |
| Russian Science Foundation | 21-64-00006 | Olga A Dontsova |
| Russian Science Foundation | 19-14-00115 | Vladimir I Polshakov |

The funders had no role in study design, data collection and interpretation, or the decision to submit the work for publication.

### Author contributions

Alexander N Malyavko, Conceptualization, Formal analysis, Funding acquisition, Investigation, Methodology, Validation, Visualization, Writing - original draft; Olga A Petrova, Investigation, Visualization, Writing – review and editing; Maria I Zvereva, Funding acquisition, Investigation, Project administration, Supervision, Writing – review and editing; Vladimir I Polshakov, Funding acquisition, Investigation, Visualization, Writing – review and editing; Olga A Dontsova, Conceptualization, Funding acquisition, Investigation, Project administration, Supervision, Writing – review and editing

### Author ORCIDs

Alexander N Malyavko ⓘ http://orcid.org/0000-0001-5064-2704
Vladimir I Polshakov ⓘ http://orcid.org/0000-0002-3216-5737

### Decision letter and Author response

Decision letter https://doi.org/10.7554/eLife.75010.sa1
Author response https://doi.org/10.7554/eLife.75010.sa2

## Additional files

### Supplementary files

• Supplementary file 1. Multiple alignment of Rif1 homologues from budding yeasts.

• Supplementary file 2. Multiple alignment of Rif1$_{NTE}$ regions from *H. polymoprha* DL-1 and five of its closest relatives.

- Supplementary file 3. *H. polymorpha* strains used in this study.
- Transparent reporting form

## Data availability

All data generated or analysed during this study are included in the manuscript and supporting files. Source data files have been provided for figures 1B, 1D, 1F, 1E, 2B, 3B, 3C, 3D, 3E, 4B, 4C, 4D, 4E, 4F, 5D, 5E, 5F, 5G, 5H, 5I, 5J, 6A, 6E, 6F, figure 1 - figure supplement 1B and C, figure 3 - figure supplement 1, figure 4 - figure supplement 1, figure 4 - figure supplement 2B, C and D, and for figure 5 - figure supplement 2A, B, C, D, E and F.

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
