## [Editor Report]

This paper presents a biochemical and functional analysis of the telomere-binding protein Rif1 (Rap1-interacting factor 1) in a budding yeast species (Hansenulapolymorpha; Hp) distantly related to the well-studied *Saccharomyces cerevisiae*. They present convincing evidence for a DNA-binding function of HpRif1 encoded in a conserved N-terminal domain predicted to be intrinsically disordered, as well as a region in the N-terminus that promotes interaction with the Hp Ku heterodimer. They also identify and characterize a functional interaction of Hp Rif1 with the Stn1 protein. Given these findings and the apparent absence of strong interaction with Rap1 in Hp, the authors provide an emerging picture of Hp Rif1 function that differs from that seen in *Saccharomyces cerevisiae*, consistent with the general picture of the rapid evolution of telomeric proteins.

---

## [Decision Letter]

**Decision letter after peer review:**

[Editors’ note: the authors submitted for reconsideration following the decision after peer review. What follows is the decision letter after the first round of review.]

Thank you for submitting your work entitled "Telomere length regulation by Rif1 protein from *Hansenula polymorpha*" for consideration by *eLife*. Your article has been reviewed by 3 peer reviewers, including Raymund J Wellinger as the Reviewing Editor and Reviewer #1, and the evaluation has been overseen a Senior Editor. The following individual involved in review of your submission has agreed to reveal their identity: David Shore (Reviewer #2).

We are sorry to say that, after consultation with the reviewers, we have decided that your work will not be considered further for publication by *eLife*.

Overall, the reviewers were impressed by the quality of the data and appreciated the attention to an important topic that is relevant for genome stability mechanisms in many eukaryotes, including mammals. As such, your interpretation of the data would impy an important deviation from the concepts established for Rif1 function in in budding yeast. However, several aspects of the hypotheses put forward are not enough supported by actual experiments and in the present form, the ideas therefore remain too speculative for publication in *eLife*. Specifically, reviewers point out that a very important part of the conclusions hinge on negative results of Y2H approaches and an inability to pinpoint a HpRif1 interaction domain on HpRap1. These results are viewed as unconvincing, and this conclusion will need substantial experimental work to be established. Furthermore, the point mutations in the NTE, when analyzed in the FL length protein, appear to retain a significant level of function (see Figure 4F, compare mutants vs rif1 deletion). It therefore remains unclear as to how and what the implications of those a.a. are on HpRfi1 function on telomeres. Finally, in order to broaden the significance and interest for the findings to other organisms and fields, it will be crucial to establish that some of the proposed new interactions and/or recruitment modes be found to be conserved in divergent species (bakers' yeast; mouse; human). As is, this discussed link remains too speculative. The reviewers nevertheless agreed that this aspect of the work too could become part of a highly significant and important contribution, if solidly supported by experimentation. Yet, given that this again implies huge additional efforts, it was convened that this is beyond a revision that could be completed within a reasonable timeframe.

*Reviewer #1:*

The Dontsova-Malyavko labs have been investigating telomere and telomerase biology in the thermotolerant budding yeast H. polymorpha since many years. While the findings in general parallel those in the classical budding yeast (*S. cerevisiae*), there already are a few interesting differences between S.c. and H.p. and for some characteristics, H.p. telomere biology appears more like other systems. In this manuscript, they describe their efforts to elucidate the function of the HpRif1 protein. This protein was originally discovered in budding yeast as a factor associating with the telomeric Rap1 protein and acting as a negative regulator for telomere lengthening by telomerase. More recently, a homologue of the protein was described in mammalian cells, including humans, and this protein was shown to be very important for DSB protection and repair choice. Furthermore, the yeast protein also englobes a regulatory function in DNA replication initiation. Hence, Rif1 is a large protein with a number of seemingly disparate functions in the studied systems.

The results of this paper first describe the identification of the HpRif1 and that it is indeed involved in telomere length restriction, as in S.c. The data show that a DBD comprised of two sequence blocks in the N-terminal part can mediate direct telomeric DNA binding in vitro but the consequences of mutating those blocks in the full-length (FL) protein in vivo remains unclear. Several approaches allowed to predict that this N-terminal extension (NTE) structurally appears unique without predictable motifs (unordered domain). Yeast 2-hybrid analyses, ChIPs and co-IP experiments suggest that the HpRif1 protein interacts with two other factors, namely HpKu and HpStn1. Both of these have important functions at telomeres as well as other sites in the genome. There is some evidence that the HpRif1 – HpKu80 interaction occurs via a specific site in the NTE again, but there appear not to be any in vivo consequences of mutating that site in the FL HpRif1 protein, which renders it difficult to rationalise an interaction via this site. On the other hand, in the absence of HpKu, HpRif1 localisation to telomeres appears significantly reduced (4-fold down) and as opposed to the situation in S.c., an absence of HpRif1 appears not to suppress a lack of Ku in terms of telomeric phenotypes. The interaction of HpRif1 with HpStn1 is documented by 2 hybrid and co-IP experiments. Furthermore, the in vivo recruitment of HpStn1 to telomeres appears dependent on HpRif1 (by ChIP). Finally, deletions of the HpStn1 C-terminal domain cause telomere length dysregulation and a loss of HpCdc13 interaction, as in S.c.

Altogether, these results suggest an intricate network of interactions between the HpRif1, HpStn1 and the HpKu proteins. These plus the NTE DNA binding domain would determine recruitment of the proteins to telomeres. In particular, the HpRif1 would be recruited entirely in a HpRap1 independent fashion, which would significantly deviate from other yeast systems. This latter point is not solidly investigated though and needs a much more thorough study. Furthermore, it is said that the HpKu does not interact with its telomerase RNA, yet I could not find any convincing experimental verification of that (RIP or so). Absence of a substructure in a secondary prediction model for the RNA is not sufficient for this point. Most importantly, the results around the HpRif1, HpKu and HpStn1 interactions are the most significant steps forward for our conceptual understanding, but certain details leave alternative explanations possible and those results do need more support for increased robustness of the conclusions.

This reviewer therefore thinks that while there is potential in these results, the core results and most exciting interpretations need to be clarified or more strongly supported by experiments.

Specifically:

1) Telomeric phenotypes in cells that lack HpKu are incompletely documented and not really clear to me. Are the telomeres shorter in the mutants and, if yes, by how much? For the telomere Southern blots, the ref Sohn et al. 1999 should be used as that paper is the one to give the details of telomeric sequence organization in H.p.

2) There is a bit of a disconnect in Figure 4. While the FL Rif1 protein with the 4A and the 8A mutations appears not to bind to telomeres at all anymore (by ChIP, Figure 4D); the telomeric phenotype conferred by these mutations is mild at best (Figure 4F). At any rate, a complete deletion of SpRif1 causes a much more dramatic telomere elongation phenotype. To this reviewer, this suggests that the identified NTE DNA binding domain is only part of the story and the conclusions must be attenuated to reflect that fact.

3) Similar to pt 2, the identified Ku-interaction domain on the NTE appears not important when put in the context of the FL Rif1 (Figure 5E. 5F). Furthermore, the co-IP experiment in Figure 5I shows a Ku80 band in lane 5; but according to my reading, there should not be one. Finally, the Southern blot (Figure 5J) may show an absence of telomere lengthening in the rfi1 ku80 double mutant, but the differences appear small and in the absence of a quantification of telomere lengths, the conclusions drawn are tantalizing, but not convincing.

4) All two hybrid analyses with the FL Rif1 or subfragment constructs were made with the Rif1 protein (or variants) fused to the AD. Does the reverse order 2 hybrid (Rif1 fused to GBD) yield the same results?

5) In strains lacking Rif1, does Ku still bind to telomeres (by ChIP)?

6) In the absence of Ku, Rif1 binding to telomeres appears down, but is this just an effect of the shorter telomeres? Thus, in another short telomere mutant (not Ku), is Rif1 bound at the wt level or reduced too? In S.c., telomere length does correlate with Rif1 binding, which would easily explain the result in Figure 5D.

7) For Figure 6D, it must be mentioned that such C-terminal truncations with a loss of Cdc13 interaction have been shown to cause exactly the same phenotype in S.c. (Petreaca et al.; NCB 2007). Whether or not this has anything to do with a potential Rif1 interaction remains to be determined. From the Southern blot in Figure 6D, it would also appear that the FLAG epitope on Stn1 causes some impairment, given the longer telomeres in that strain.

*Reviewer #2:*

Malyavko et al. present a biochemical and functional analysis of the telomere regulator Rif1 in a budding yeast species (Hansenula polymorpha; Hp) distantly related to the well-studied *Saccharomyces cerevisiae* (Sc), where Rif1 was first identified and is best characterized. They present convincing evidence for a DNA-binding function of HpRif1 encoded in a conserved N-terminal domain predicted to be intrinsically disordered, as well as a region in the N-terminus that promotes interaction with the Hp Ku heterodimer. Based upon these findings and a series of negative results related to known functional domains and interacting partners of ScRif1, they propose that Hp and its close relatives have evolved a strikingly different mode of recruitment of Rif1 at telomeres compared to Sc. The novel interactions reported are of interest, but the data so far presented indicate that they make only a minor contribution to Rif1's overall role in telomere length regulation. In this regard, there is concern that the authors place too much weight on a single negative result related to the possible role of Rap1 in the recruitment of Rif1 in Hp, an issue that needs to be more carefully addressed.

As a whole, the experiments presented here are well designed and executed and the data are clearly presented and appear to be of high quality. This is a substantial piece of experimental work that will be of high interest to those in the telomere field. My concerns (detailed below) are primarily related to interpretation. Put briefly, I am worried that the authors have missed a potentially important role for HpRap1 in HpRif1 binding and telomere length regulation simply because they failed to detect a Y2H interaction between the two. This is a negative result that carries little weight. The novel findings reported here, as the authors acknowledge, still provide an incomplete picture of HpRif1 telomere recruitment and function. As such, the present manuscript, though interesting, is still rather preliminary. Specific points and questions are as follows:

1. The conclusion that the HpRif1 NTE is required for telomere recruitment is questionable. The data clearly indicate an important role for the two basic motifs in DNA binding both in vitro and in vivo, but the functional assay of the mutants (the telomere blot in Figure 4F) clearly shows that the mutants retain a significant telomere length regulation function compared to a rif1Δ strain. (Indeed, the mutants display more heterogeneity in the lower telomere-specific (?) bands compared to wild-type but nothing like the essentially uniform and pronounced extension of the deletion strain). In any event, these data suggest that Rif1 is still functioning at telomeres in the K/R cluster mutant strains. The simplest explanation of these observations is that the mutant proteins are still recruited to telomeres, even though the ChIP assay no longer reports this binding. This may be due either to a more transient binding, not capture by ChIP, or to a decrease in crosslinking efficiency provoked by the mutations themselves. In short, the effect of this putative DNA-binding domain is rather minor.

2. The authors have somewhat hastily dismissed the notion that the ScRif1 NTD homology region in HpRif1 (which is relatively conserved) might be involved in DNA binding. As best I can tell, they may have simply failed to express this part of HpRif1 in a soluble or properly folded form that would allow for biochemical analysis. This point needs to be clarified and if possible, resolved experimentally. The authors should also comment on the conservation of amino acid residues shown to be involved in ScRif1 DNA binding by Mattarocci et al. (2017). If present in HpRif, these residues should be mutated and tested in functional assays.

3. Identification and analysis of the novel Ku-binding domain in HpRif1 suffers from several of the problems cited above concerning the DNA-binding domain. In this case, though, only the Y2H analysis indicates a loss of binding for the mutants that were generated, at least in the context of full-length HpRif1. The failure of Rif1NTE1-264 fragments carrying these mutations to bind (as measured by ChIP) does support an in vivo role for the Ku interaction but again highlights a relatively minor functional role, at least concerning telomere length regulation.

4. The authors should test the functional redundancy of DNA- and Ku-binding by generating and characterizing double mutants of the relevant motifs.

5. The authors' conclusion that HpRap1 plays no role in HpRif1 telomere recruitment, if true, is a very significant finding given our present understanding of telomere length regulation. The so-called "protein counting" model posits that telomere length is "read out" by a system that is sensitive to the number of bound Rap1 molecules. If HpRap1 plays no role in recruiting HpRif1, it is unclear how Rif1 would operate to control telomere length. The authors need to at least discuss this issue.

6. Regarding the above (HpRap1's role in telomere length regulation), the authors should discuss HpRap1's homology with ScRap1 and if not already done, make mutations in the C-terminal domain of HpRap1 that might be likely to play a role in telomere length regulation. Surprisingly, I find no discussion of this issue, apart from their claim that a Rap1-binding motif (RBM) cannot be identified in HpRif1. In Sc, the RBM interacts with a groove in the C-terminal domain of ScRap, and the same groove is used by the silencing protein Sir3 for ScRap1 binding. Furthermore, even human TRF2 interacts with human Rap1 through a similar mechanism (see Chen et al. 2011 NSMB). I would thus be surprised if something similar does not occur in Hp, though it might be difficult to uncover by sequence homology searches. This is something that needs to be addressed experimentally.

*Reviewer #3:*

Rif1 has been implicated in DNA replication and DNA damage repair in mammals and yeast. In *Saccharomyces cerevisiae* (and other yeast species) it was also shown to negatively regulate telomere length by inhibiting telomerase action. The authors aimed at understanding the interactions and function of Rif1 in a distant budding yeast species, Hansenula polymorpha. The authors show that deleting Rif1 in H. polymorpha indeed causes telomere elongation, confirming the conserved function of Rif1 as a negative regulator of telomerase. However, while *S. cerevisiae* Rif1 is recruited to telomeres by Rap1, here the authors describe two other types of interactions that contribute to the recruitment of H. polymorpha Rif1 to telomeres: a direct interaction of its unstructured N-terminal domain with the telomeric DNA and another interaction with the Ku heterodimer, also mediated (at least in part) by the N-terminal domain of Rif1. The authors also show that H. polymorpha Rif1 recruits Stn1 to telomeres, and Stn1 negatively regulates telomerase, as shown in other yeasts. Altogether, the authors portray a network of interactions at the telomere, which are different from what has been reported in other yeasts, but they are also important for the regulation of telomerase action at the telomere. Possibly, some of these interactions also take place at the telomeres of *S. cerevisiae* and other yeasts, and they have not been detected because of redundancy. Interestingly, disrupting these interactions results in effects on telomere length that are not readily explained by a simple recruitment function, suggesting that the roles of these interactions are more complex and/or other unknown interactions are yet to be discovered. The experiments are well designed and conducted with the proper controls, the results are convincing and the manuscript at large is clearly written. The manuscript would be interesting to readers in the telomere field as they portray a more complex network of interactions taking place at the telomere, which is likely to occur also in other species. In my opinion, however, this work still leaves open questions. It could have a stronger impact, especially outside the yeast telomere field, if the mechanism of telomerase regulation was explored in more detail or if the interactions found in H. polymorpha were shown to be conserved also in other species.

1. I believe that the introduction would benefit from mentioning what is known about other telomeric proteins in H. polymorpha, such as the CST complex and its interactions with telomerase, Rif2 homolog (if exists). As well as mentioning the more recent papers about Rif1 from the Greider group.

2. The authors did not find the Rap1 interacting domain within Rif1, but have not excluded that Rif1 interacts with Rap1, perhaps through a different domain. Experiments testing this potential interactions would strengthen the conclusions.

3. Figure 1C: It appears that TER deletion is lethal. Is it indeed the case? Are there no survivors by the alternative recombination-based mechanism?

4. Figure 1B,E: some of the lanes display altered pattern of telomeric restriction fragments. Do the authors consider how are they formed? Is it sub-telomeric recombination, and is it related to Rif1?

5. Figure 4: The Rif1 8A mutant abolished the interaction with the telomeric DNA, and yet it did not cause a significant telomere elongation. This is puzzling. Is this interaction dispensable for telomere length regulation? Can Rif1 inhibit telomerase if only transiently localized to telomeres? Or even without binding to telomeres?

6. Lines 258-260: Could the reduced Rif1 association with telomeres of the Ku deletion strains be the result of the shorter telomeres, rather than the loss of interaction with Ku? The explanation given in lines 265-269 is not clear.

7. Figure 6F: The ChIP is not clearly described. Is Stn1-HA immunoprecipitated here? Please indicate the target of the pull-down.

8. Figure 6D: Could be interesting to test double mutants such as Stn1delta C with ku or rif1.

9. Can the authors draw a schematic model to illustrate the identified interactions and how they might regulate telomerase?

10. In several places 'the' or 'a' are missing (e.g., line 48 – THE C-terminal portion, line 212, line 300…).

11. Line 51: Does Rif1 inhibit or mediate telomere silencing?

12. Figure 1A, Supplementary file 1, give reference to T-Coffee.

13. Line 155: 'is' is missing.

14. The figure legends could benefit from a better description of the figures, for example, In Figure 2A, what is the difference between the lines MetaDisorder, MetaDisorderMD and MetaDisorderMD2?

15. Line 179: May be remove 'and' and start a new sentence?

16. Line 190 and Figure 3E: When titrating a C4 competitor in a reaction with a G4 probe, first it eliminates the single stranded G4 by generating a double stranded GC4 probe. Then the excess C4 competes with the GC4 probe on the binding of the protein. Thus the results are consistent with a preference for the single stranded G4 as seen in 3D.

17. Line 371: Not in all yeasts the role of Ku is conserved (e.g. K. lactis). I'm not convinced that the binding of Ku to telomerase is conserved and we can assume that it occurs also in H. polymorpha.

---

## [Author Response]

[Editors’ note: the authors resubmitted a revised version of the paper for consideration. What follows is the authors’ response to the first round of review.]

Reviewer #1:The Dontsova-Malyavko labs have been investigating telomere and telomerase biology in the thermotolerant budding yeast H. polymorpha since many years. While the findings in general parallel those in the classical budding yeast (*S. cerevisiae*), there already are a few interesting differences between S.c. and H.p. and for some characteristics, H.p. telomere biology appears more like other systems. In this manuscript, they describe their efforts to elucidate the function of the HpRif1 protein. This protein was originally discovered in budding yeast as a factor associating with the telomeric Rap1 protein and acting as a negative regulator for telomere lengthening by telomerase. More recently, a homologue of the protein was described in mammalian cells, including humans, and this protein was shown to be very important for DSB protection and repair choice. Furthermore, the yeast protein also englobes a regulatory function in DNA replication initiation. Hence, Rif1 is a large protein with a number of seemingly disparate functions in the studied systems.The results of this paper first describe the identification of the HpRif1 and that it is indeed involved in telomere length restriction, as in S.c. The data show that a DBD comprised of two sequence blocks in the N-terminal part can mediate direct telomeric DNA binding in vitro but the consequences of mutating those blocks in the full-length (FL) protein in vivo remains unclear. Several approaches allowed to predict that this N-terminal extension (NTE) structurally appears unique without predictable motifs (unordered domain). Yeast 2-hybrid analyses, ChIPs and co-IP experiments suggest that the HpRif1 protein interacts with two other factors, namely HpKu and HpStn1. Both of these have important functions at telomeres as well as other sites in the genome. There is some evidence that the HpRif1 – HpKu80 interaction occurs via a specific site in the NTE again, but there appear not to be any in vivo consequences of mutating that site in the FL HpRif1 protein, which renders it difficult to rationalise an interaction via this site. On the other hand, in the absence of HpKu, HpRif1 localisation to telomeres appears significantly reduced (4-fold down) and as opposed to the situation in S.c., an absence of HpRif1 appears not to suppress a lack of Ku in terms of telomeric phenotypes. The interaction of HpRif1 with HpStn1 is documented by 2 hybrid and co-IP experiments. Furthermore, the in vivo recruitment of HpStn1 to telomeres appears dependent on HpRif1 (by ChIP). Finally, deletions of the HpStn1 C-terminal domain cause telomere length dysregulation and a loss of HpCdc13 interaction, as in S.c.Altogether, these results suggest an intricate network of interactions between the HpRif1, HpStn1 and the HpKu proteins. These plus the NTE DNA binding domain would determine recruitment of the proteins to telomeres. In particular, the HpRif1 would be recruited entirely in a HpRap1 independent fashion, which would significantly deviate from other yeast systems. This latter point is not solidly investigated though and needs a much more thorough study. Furthermore, it is said that the HpKu does not interact with its telomerase RNA, yet I could not find any convincing experimental verification of that (RIP or so). Absence of a substructure in a secondary prediction model for the RNA is not sufficient for this point. Most importantly, the results around the HpRif1, HpKu and HpStn1 interactions are the most significant steps forward for our conceptual understanding, but certain details leave alternative explanations possible and those results do need more support for increased robustness of the conclusions.This reviewer therefore thinks that while there is potential in these results, the core results and most exciting interpretations need to be clarified or more strongly supported by experiments.

We thank the reviewer for the appreciation of our work and for the careful reading of the manuscript. To obtain new data on the potential HpRif1-HpRap1 relationship, we constructed the intTEL18 strain of H. polymorpha by inserting 18 telomeric repeats upstream of the WSC3 gene on the chromosome I (Figure 1 —figure supplement 1A). As expected, Rap1B binds telomeric repeats with similar efficiency at the internal locus or at the chromosome end (only ~2-fold difference between WSC3 and TEL signals for the intTEL18 strain, Figure 1 —figure supplement 1B). On the contrary, Rif1 association with the internal telomeric repeats is weak and close to the background levels (ChIP WSC3 signal is ~8-fold lower than TEL, Figure 1 —figure supplement 1C). We believe that this experiment together with our previous results (the phenotypic discrepancy between Rap1 and Rif1 mutants, the absence of the Y2H interaction) allows us to conclude that “it is highly unlikely that Rif1 binding to Rap1 is the major mechanism of Rif1 telomeric recruitment in H. polymorpha in a manner it is described for its *S. cerevisiae* homologues.”

Specifically:1) Telomeric phenotypes in cells that lack HpKu are incompletely documented and not really clear to me. Are the telomeres shorter in the mutants and, if yes, by how much? For the telomere Southern blots, the ref Sohn et al. 1999 should be used as that paper is the one to give the details of telomeric sequence organization in H.p.

To provide a quantitative estimate of the telomere length changes observed in Ku mutants, we calculated the lengths of the brightest TRFs on the Southern blots and report them in Figures and in the text. Since it is not clear what is the exact length of the subtelomere region, we used the value ~160 bp (~20 telomeric repeats) as an estimate of the mean WT telomere length (18-23 telomeric repeats reported in Sohn et al. 1999). We found that Ku loss leads to telomere shortening (~25-50% reduction in telomere length, Figure 5 —figure supplement 2B, Figure 5J).

2) There is a bit of a disconnect in Figure 4. While the FL Rif1 protein with the 4A and the 8A mutations appears not to bind to telomeres at all anymore (by ChIP, Figure 4D); the telomeric phenotype conferred by these mutations is mild at best (Figure 4F). At any rate, a complete deletion of SpRif1 causes a much more dramatic telomere elongation phenotype. To this reviewer, this suggests that the identified NTE DNA binding domain is only part of the story and the conclusions must be attenuated to reflect that fact.

We have changed our conclusions accordingly (see for example, page 22 lines 421-423; page 25 lines 485-486).

3) Similar to pt 2, the identified Ku-interaction domain on the NTE appears not important when put in the context of the FL Rif1 (Figure 5E. 5F).

We agree and we state that “The Rif1_KBM_ may thus serve only as an auxiliary module” (please see pages 22-23 lines 430-434 of the Discussion).

Furthermore, the co-IP experiment in Figure 5I shows a Ku80 band in lane 5; but according to my reading, there should not be one.

Indeed, in the old version of the Figure 5I (Author response image 1) the background level of Ku80-HA associated with the anti-Flag resin is higher for the ∆ku70 strain (lanes 5 and 6) compared to the KU70 strain (lane 7). Our intention was to demonstrate that Ku80-HA is no longer associated with Rif1-Flag (relatively to the “no Flag tag” control, since lanes 5 and 6 are the same). We do not know the reason for this higher non-specific binding; however, we understand that it may be confusing and removed the experiment with the ∆ku70 strain. We note that the pull down from the KU70 strain does support our conclusion that Rif1 interacts with Ku80 in H. polymorpha cells (the Ku80 signal in lane 8 is much higher than in lane 7, and lanes 5,6) and we show it in the new version of the manuscript (new Figure 5D).

**Author response image 1. sa2fig1:** Co-IP analysis. Rif1-Flag protein was immunoprecipitated from the indicated strains on the anti-Flag resin. The amount of tagged proteins in whole cell extracts (WCE) and IP samples (IP) was monitored by Western blot (WB) using anti-Flag and anti-HA antibodies. The IP experiment was performed in the presence of benzonase nuclease.

Finally, the Southern blot (Figure 5J) may show an absence of telomere lengthening in the rfi1 ku80 double mutant, but the differences appear small and in the absence of a quantification of telomere lengths, the conclusions drawn are tantalizing, but not convincing.

We quantified the lengths of the brightest TRFs on the Southern blot in Figure 5J and found that the length increase upon the Rif1 loss in ∆ku80 is ~10 bp (399±1 bp for RIF1∆ku80 vs 410±11 bp for ∆rif1∆ku80). This constitutes ~13% (10/80) of the ∆ku80 telomere length (KU80 TRF is ~480 bp, ∆ku80 TRF is ~400 bp; WT telomere tract is ~160 bp, therefore in ∆ku80 telomere tract is ~80 bp). We did not quantify the TRFs in the rif1 mutants, because of the high heterogeneity and it is not clear which TRF corresponds to the brightest in the WT strain. However, from the Southern blot in Figure 5J it can be seen that the mean length of the brightest TRFs is close to 750 bp. Therefore, the length increase upon the Rif1 loss in the KU80 strain should be ~270 bp, which is more than 100% of the WT telomere length. Thus, we believe that the following statement is justified: “RIF1 deletion in a ∆ku80 background does not lead to strong telomere elongation compared to the parental ∆ku80 strain, suggesting that inhibition of telomerase by Rif1 is attenuated in the absence of Ku (Figure 5J).” (lines 323-325).

4) All two hybrid analyses with the FL Rif1 or subfragment constructs were made with the Rif1 protein (or variants) fused to the AD. Does the reverse order 2 hybrid (Rif1 fused to GBD) yield the same results?

We chose not to perform the reverse order 2 hybrid experiments because we believe that our main conclusions based on the Y2H assay are corroborated by more informative complementary approaches (Co-IPs, ChIPs and Southern blots).

5) In strains lacking Rif1, does Ku still bind to telomeres (by ChIP)?

Yes, Ku80-HA binds telomeres in the ∆rif1 strain, with only ~2-fold reduced efficiency compared to the RIF1 strain (Figure 5 —figure supplement 2C).

6) In the absence of Ku, Rif1 binding to telomeres appears down, but is this just an effect of the shorter telomeres? Thus, in another short telomere mutant (not Ku), is Rif1 bound at the wt level or reduced too? In S.c., telomere length does correlate with Rif1 binding, which would easily explain the result in Figure 5D.

To verify that Rif1 telomere localization defect upon Ku loss is not simply a consequence of the telomere shortening, we measured Rif1 telomere binding in a strain lacking telomerase RNA (new Figure 5 —figure supplement 2D, E). TER knock-out resulted in ~40% reduction in telomere length, whereas Rif1-HA telomere occupancy diminished only ~2-fold (new Figure 5 —figure supplement 2D, E); contrasting to ~4-fold Rif1 ChIP signal drop in case of Ku mutants (new Figure 5E).

7) For Figure 6D, it must be mentioned that such C-terminal truncations with a loss of Cdc13 interaction have been shown to cause exactly the same phenotype in S.c. (Petreaca et al.; NCB 2007). Whether or not this has anything to do with a potential Rif1 interaction remains to be determined. From the Southern blot in Figure 6D, it would also appear that the FLAG epitope on Stn1 causes some impairment, given the longer telomeres in that strain.

We included this and two other references with descriptions that Stn1 C-terminal truncations have been shown to cause exactly the same phenotype in *S. cerevisiae*. (lines 362-363). We agree that FLAG epitope on Stn1 causes some functional impairment, however we could not see how this fact changes any of our conclusions.

Reviewer #2:Malyavko et al. present a biochemical and functional analysis of the telomere regulator Rif1 in a budding yeast species (Hansenula polymorpha; Hp) distantly related to the well-studied *Saccharomyces cerevisiae* (Sc), where Rif1 was first identified and is best characterized. They present convincing evidence for a DNA-binding function of HpRif1 encoded in a conserved N-terminal domain predicted to be intrinsically disordered, as well as a region in the N-terminus that promotes interaction with the Hp Ku heterodimer. Based upon these findings and a series of negative results related to known functional domains and interacting partners of ScRif1, they propose that Hp and its close relatives have evolved a strikingly different mode of recruitment of Rif1 at telomeres compared to Sc. The novel interactions reported are of interest, but the data so far presented indicate that they make only a minor contribution to Rif1's overall role in telomere length regulation. In this regard, there is concern that the authors place too much weight on a single negative result related to the possible role of Rap1 in the recruitment of Rif1 in Hp, an issue that needs to be more carefully addressed.As a whole, the experiments presented here are well designed and executed and the data are clearly presented and appear to be of high quality. This is a substantial piece of experimental work that will be of high interest to those in the telomere field. My concerns (detailed below) are primarily related to interpretation. Put briefly, I am worried that the authors have missed a potentially important role for HpRap1 in HpRif1 binding and telomere length regulation simply because they failed to detect a Y2H interaction between the two. This is a negative result that carries little weight. The novel findings reported here, as the authors acknowledge, still provide an incomplete picture of HpRif1 telomere recruitment and function. As such, the present manuscript, though interesting, is still rather preliminary. Specific points and questions are as follows:

We thank the reviewer for the time and effort spent on reading of our manuscript and for the valuable comments and critique. We wanted to stress that our conclusion on the absence of the major role for HpRap1 in HpRif1 recruitment stems mainly not from the negative results of the Y2H experiments but from the different phenotypes of the strains with HpRap1 C-truncations and RIF1 knock-out (please see the detailed response to the point #6). We understand that in our previous version of the study this was not discussed properly and tried to extend the description of our previous results with HpRap1 C-truncations (page 21 lines 381-393). In addition, we included the results with the newly created intTEL18 strain containing an array of telomeric repeats at the internal WSC3 locus (Figure 1 —figure supplement 1A). As expected, HpRap1B binds telomeric repeats with similar efficiency at the internal locus or at the chromosome end (only ~2-fold difference between WSC3 and TEL signals for the intTEL18 strain, Figure 1 —figure supplement 1B). On the contrary, HpRif1 association with the internal telomeric repeats is weak and close to the background levels (ChIP WSC3 signal is ~8-fold lower than TEL, Figure 1 —figure supplement 1C). We strongly believe that these results (together with the Y2H assay) make enough evidence to conclude that “it is highly unlikely that Rif1 binding to Rap1 is the major mechanism of Rif1 telomeric recruitment in H. polymorpha in a manner it is described for its *S. cerevisiae* homologues.”

1. The conclusion that the HpRif1 NTE is required for telomere recruitment is questionable. The data clearly indicate an important role for the two basic motifs in DNA binding both in vitro and in vivo, but the functional assay of the mutants (the telomere blot in Figure 4F) clearly shows that the mutants retain a significant telomere length regulation function compared to a rif1Δ strain. (Indeed, the mutants display more heterogeneity in the lower telomere-specific (?) bands compared to wild-type but nothing like the essentially uniform and pronounced extension of the deletion strain). In any event, these data suggest that Rif1 is still functioning at telomeres in the K/R cluster mutant strains. The simplest explanation of these observations is that the mutant proteins are still recruited to telomeres, even though the ChIP assay no longer reports this binding. This may be due either to a more transient binding, not capture by ChIP, or to a decrease in crosslinking efficiency provoked by the mutations themselves. In short, the effect of this putative DNA-binding domain is rather minor.

We agree with the reviewer. HpRif1 mutants with K/R cluster mutations (or even lacking the entire NTE) retain large portion of Rif1’s functional potential. We discuss this in the manuscript (see, for example, page 22 lines 421-427). Indeed, the contribution of the NTE region to telomere length regulation is relatively small, although it does appear to be important for “full” inhibition of telomerase by HpRif1. We attenuated our conclusions throughout the text to reflect that fact. We now state: “We demonstrated that the N-terminal extension (NTE) of Rif1 is an accessory domain partially promoting Rif1 telomeric localization in H. polymorpha.” (page 25, lines 485-486).

2. The authors have somewhat hastily dismissed the notion that the ScRif1 NTD homology region in HpRif1 (which is relatively conserved) might be involved in DNA binding. As best I can tell, they may have simply failed to express this part of HpRif1 in a soluble or properly folded form that would allow for biochemical analysis. This point needs to be clarified and if possible, resolved experimentally. The authors should also comment on the conservation of amino acid residues shown to be involved in ScRif1 DNA binding by Mattarocci et al. (2017). If present in HpRif, these residues should be mutated and tested in functional assays.

We agree with the reviewer and discuss the limitations of our approach and the potential conservation of the DNA binding by HpRif1 NTD (page 13 lines 262-265, page 22 lines 413415; page 22 lines 426-427; page 25 lines 490-493). We tried to find whether DNA-contacting residues from ScRif1 NTD are conserved in HpRif1 and present the results in this version of the manuscript. According to the published ScRif1_NTD_/DNA structure, 19 charged residues have the potential to contact the DNA backbone (Mattarocci et al., 2017). Of these 19, only four are conserved in H. polymorpha Rif1 (new Figure 4 —figure supplement 2A, Supplementary file 1).

Double mutant HpRif1_K658E/K666E_ is expressed at considerably lower levels than WT HpRif1, indicating that residues K658 and K666 are important for protein stability (new Figure 4 —figure supplement 2B). This would complicate interpretation of the functional assays and these residues were excluded from further analysis. Substitution of the other two conserved amino acids (K504 and R539) for glutamines does not lead to detectable changes in Rif1 telomere occupancy or telomere length (Figure 4 —figure supplement 2C, D). These results suggest that the HpRif1 might lack the DNA-binding activity described for ScRif1. However, since the telomere-related proteins (especially in yeast) are highly evolvable we still do not dismiss the possibility of DNA binding by HpRif1 NTD (page 22, lines 418-420).

3. Identification and analysis of the novel Ku-binding domain in HpRif1 suffers from several of the problems cited above concerning the DNA-binding domain. In this case, though, only the Y2H analysis indicates a loss of binding for the mutants that were generated, at least in the context of full-length HpRif1. The failure of Rif1NTE1-264 fragments carrying these mutations to bind (as measured by ChIP) does support an in vivo role for the Ku interaction but again highlights a relatively minor functional role, at least concerning telomere length regulation.

Indeed, our experiments indicate that other residues within HpRif1 (its NTE-less/265-1521 portion) make crucial contacts with Ku in vivo. The inability to detect the interaction between Ku80 and Rif1^264-1521^ in the Y2H experiment might even suggest that this interaction requires Ku as a whole, or protein modifications on Ku/Rif1 or other Ku/Rif1-bound proteins. Unfortunately, we could not provide more information on this issue. We would like to note, however, that we discuss that in the manuscript (page 16 lines 307-308; page 23, lines 430-436).

4. The authors should test the functional redundancy of DNA- and Ku-binding by generating and characterizing double mutants of the relevant motifs.

We thank the reviewer for an interesting suggestion, however we believe that the outcome of the experiments would be similar to the Rif1^∆NTE^ strain (since both DNA- and Ku-binding motifs are located within the NTE region). Rif1^∆NTE^ mutant is “not present” at telomeres according to ChIP and has moderately elongated telomeres (Figure 1D, E). The DNA-binding mutant Rif1^8A^ has the identical phenotype (Figure 4D, F).

5. The authors' conclusion that HpRap1 plays no role in HpRif1 telomere recruitment, if true, is a very significant finding given our present understanding of telomere length regulation. The so-called "protein counting" model posits that telomere length is "read out" by a system that is sensitive to the number of bound Rap1 molecules. If HpRap1 plays no role in recruiting HpRif1, it is unclear how Rif1 would operate to control telomere length. The authors need to at least discuss this issue.

Following the suggestion by the reviewer, we introduced the paragraph on this matter in the Discussion section (page 23 lines 437-448). At this point it is not obvious how mechanistically Rif1 recruitment could be linked to telomere length. HpRif1 appears to have only limited ability to bind double-stranded telomeric DNA in vivo (Figure 1 —figure supplement 1). We also would like to note that ScRif1^NTD^ interacts with DNA in a Rap1-independent (and telomere length independent) manner. This suggests that Rif1 homologues operate (at least partially) in a telomere length-independent way.

6. Regarding the above (HpRap1's role in telomere length regulation), the authors should discuss HpRap1's homology with ScRap1 and if not already done, make mutations in the C-terminal domain of HpRap1 that might be likely to play a role in telomere length regulation. Surprisingly, I find no discussion of this issue, apart from their claim that a Rap1-binding motif (RBM) cannot be identified in HpRif1. In Sc, the RBM interacts with a groove in the C-terminal domain of ScRap, and the same groove is used by the silencing protein Sir3 for ScRap1 binding. Furthermore, even human TRF2 interacts with human Rap1 through a similar mechanism (see Chen et al. 2011 NSMB). I would thus be surprised if something similar does not occur in Hp, though it might be difficult to uncover by sequence homology searches. This is something that needs to be addressed experimentally.

The functional characterization of HpRap1 and the homology with ScRap1 was the focus of our earlier study (Malyavko et. al 2019, Sci Rep), and the experiments therein are indeed useful for rationalizing potential Rif1-Rap1 interaction. We discovered that H. polymorpha has two Rap1 homologues and deletion of the entire RCT from one of them (Rap1A) has no effect on the telomere length, consistent with its lack of preference for the telomeric DNA sequence. RCT deletion from Rap1B (the major dsTBP in H. polymorpha) was found to be impossible (presumably lethal). Removal of the 50 a.a. from its C-terminus was found tolerated by cells but strongly (more than 10-fold) reduced the expression level of Rap1B, creating thus effectively the strain with a Rap1B knock-down. In this (B^1-526^) strain telomeres were indeed long and extremely heterogeneous, but the telomere overelongation was found to be telomerase-independent. In contrast, telomeres in the ∆rif1 strain cannot elongate upon telomerase loss (Figure 1B, C). These phenotypic differences actually formed our initial premise that Rif1 and Rap1 functions may be (largely) independent in H. polymorpha.

Reviewer #3:Rif1 has been implicated in DNA replication and DNA damage repair in mammals and yeast. In *Saccharomyces cerevisiae* (and other yeast species) it was also shown to negatively regulate telomere length by inhibiting telomerase action. The authors aimed at understanding the interactions and function of Rif1 in a distant budding yeast species, Hansenula polymorpha. The authors show that deleting Rif1 in H. polymorpha indeed causes telomere elongation, confirming the conserved function of Rif1 as a negative regulator of telomerase. However, while *S. cerevisiae* Rif1 is recruited to telomeres by Rap1, here the authors describe two other types of interactions that contribute to the recruitment of H. polymorpha Rif1 to telomeres: a direct interaction of its unstructured N-terminal domain with the telomeric DNA and another interaction with the Ku heterodimer, also mediated (at least in part) by the N-terminal domain of Rif1. The authors also show that H. polymorpha Rif1 recruits Stn1 to telomeres, and Stn1 negatively regulates telomerase, as shown in other yeasts. Altogether, the authors portray a network of interactions at the telomere, which are different from what has been reported in other yeasts, but they are also important for the regulation of telomerase action at the telomere. Possibly, some of these interactions also take place at the telomeres of *S. cerevisiae* and other yeasts, and they have not been detected because of redundancy. Interestingly, disrupting these interactions results in effects on telomere length that are not readily explained by a simple recruitment function, suggesting that the roles of these interactions are more complex and/or other unknown interactions are yet to be discovered. The experiments are well designed and conducted with the proper controls, the results are convincing and the manuscript at large is clearly written. The manuscript would be interesting to readers in the telomere field as they portray a more complex network of interactions taking place at the telomere, which is likely to occur also in other species. In my opinion, however, this work still leaves open questions. It could have a stronger impact, especially outside the yeast telomere field, if the mechanism of telomerase regulation was explored in more detail or if the interactions found in H. polymorpha were shown to be conserved also in other species.

We are grateful to the reviewer for the time and effort in evaluating our manuscript and for the valuable comments. Following your and other reviewers’ suggestions we explored the interactions between H. polymorpha telomeric proteins in more detail.

1. I believe that the introduction would benefit from mentioning what is known about other telomeric proteins in H. polymorpha, such as the CST complex and its interactions with telomerase, Rif2 homolog (if exists). As well as mentioning the more recent papers about Rif1 from the Greider group.

Thank you for the suggestion. We added a paragraph describing the telomeric proteins in H. polymorpha in the Introduction section (page 4 lines 88-97) and included the description of the results from the works Shubin and Greider, 2020; Shubin et al., 2021 (page 3 lines 55-56 and line 76)

2. The authors did not find the Rap1 interacting domain within Rif1, but have not excluded that Rif1 interacts with Rap1, perhaps through a different domain. Experiments testing this potential interactions would strengthen the conclusions.

We added new data with the strain intTEL18 containing the array of 18 telomeric repeats at the internal WSC3 locus. We found that HpRap1B (the major dsTBP in H. polymorpha) efficiently binds the internal telomeric DNA. At the same time HpRif1 binding at the internal telomeric array is strongly reduced compared to the native chromosomal end (Figure 1 —figure supplement 1B). We also note that removal of the 50 a.a. from its C-terminus was found tolerated by cells but strongly (more than 10-fold) reduced the expression level of Rap1B, creating thus effectively the strain with Rap1B knock-down (Malyavko et. al 2019, Sci Rep). In this (B^1-526^) strain telomeres were indeed long and extremely heterogeneous, but the telomere overelongation was found to be telomerase-independent. In contrast, telomeres in the ∆rif1 strain cannot elongate upon telomerase loss (Figure 1B, C). Both of these experiments suggest that Rif1 and Rap1 functions may be (largely) independent in H. polymorpha, and even if they interact through any of their domains such an interaction would play only a limited role in Rif1 recruitment.

3. Figure 1C: It appears that TER deletion is lethal. Is it indeed the case? Are there no survivors by the alternative recombination-based mechanism?

No, TER deletion is not lethal, but leads to ever shorter phenotype as in other yeast species, although ∆ter in H. polymorpha does senesce faster due to shorter telomeres (~160 bp WT length). When the ∆ter culture is propagated in liquid medium the survivors will eventually emerge (Smekalova et al., 2013). In the Figure 1C it appears lethal because we re-streaked ∆ter colonies on YPD plates (as described in Materials and methods), and it turned out that in the colonies we re-streaked no survivor cells appeared.

4. Figure 1B,E: some of the lanes display altered pattern of telomeric restriction fragments. Do the authors consider how are they formed? Is it sub-telomeric recombination, and is it related to Rif1?

This is an interesting question the answer to which we do not have yet. The altered pattern of TRFs can result both from the sub-telomeric recombination and telomere length change. It is important to note that telomeres can no longer elongate after TER deletion in the ∆rif1 strain, therefore the TRF length changes must be primarily due to telomerase activity.

5. Figure 4: The Rif1 8A mutant abolished the interaction with the telomeric DNA, and yet it did not cause a significant telomere elongation. This is puzzling. Is this interaction dispensable for telomere length regulation? Can Rif1 inhibit telomerase if only transiently localized to telomeres? Or even without binding to telomeres?

This is indeed puzzling. We do not think (neither do we state that in the text) that 8A mutation actually completely abolishes interaction with telomeres, we think it is just reduced to the levels we can’t detect by our ChIP assay. Perhaps, Rif1 8A mutant binds only at a specific cell cycle stage or its binding is transient (as we discuss that on page 22 lines 421-425). It also should be mentioned that in *S. cerevisiae* the Rif1^RBM^ mutant lacking the ability to contact Rap1 is almost completely lost from telomeres as judged by ChIP, yet retains a significant portion of its telomerase inhibitory potential (Mattarocci et al., 2017; Shi et al., 2013).

6. Lines 258-260: Could the reduced Rif1 association with telomeres of the Ku deletion strains be the result of the shorter telomeres, rather than the loss of interaction with Ku? The explanation given in lines 265-269 is not clear.

To clarify this issue, we performed the ChIP assay with the Rif1-HA∆ter strain (new Figure 5 —figure supplement 2D, E). TER knock-out resulted in ~40% reduction in telomere length, whereas Rif1-HA telomere occupancy diminished only ~2-fold (new Figure 5 —figure supplement 2D, E); contrasting to ~4-fold Rif1 ChIP signal drop in case of Ku mutants (new Figure 5E) and comparable telomere length reduction (~25-50% reduction in telomere length, Figure 5 —figure supplement 2B, Figure 5J). Thus, shorter telomeres cannot fully explain the reduction of Rif1 in the Ku deletion strains

7. Figure 6F: The ChIP is not clearly described. Is Stn1-HA immunoprecipitated here? Please indicate the target of the pull-down.

Yes, Stn1-HA is the target of IP in this experiment. We added this to the figure legend.

8. Figure 6D: Could be interesting to test double mutants such as Stn1delta C with ku or rif1.

We agree with the reviewer that these experiments could be interesting. We however chose not to perform them at this point, because we think that the results will be difficult to interpret and it is not clear how these could change the conclusions of this study. The functions of Ku, Rif1 and Stn1 only partially overlap. Stn1 has Rif1-independent functions (Stn1deltaC telomere phenotype is “stronger” than ∆rif1, Figures 1E and 6D) and can associate with telomeres with Cdc13 protein (which has strong affinity towards the telomeric G strand). We observe only limited role of Ku in Stn1 telomere recruitment (Figure 6F). Rif1 can associate with telomeres independently of Stn1 via Ku.

9. Can the authors draw a schematic model to illustrate the identified interactions and how they might regulate telomerase?

We introduced a schematic representation of the putative protein-protein interactions at H. polymorpha telomeres (Figure 6 —figure supplement 2).

10. In several places 'the' or 'a' are missing (e.g., line 48 – THE C-terminal portion, line 212, line 300…).

Thank you for pointing this out, we corrected these mistakes.

11. Line 51: Does Rif1 inhibit or mediate telomere silencing?

ScRif1 inhibit telomere silencing by competition with Sir3 for Rap1 RCT binding (as demonstrated in Shi et al., 2013).

12. Figure 1A, Supplementary file 1, give reference to T-Coffee.

We added reference to T-Coffee.

13. Line 155: 'is' is missing.

Thank you for pointing this out, we corrected the mistake.

14. The figure legends could benefit from a better description of the figures, for example, In Figure 2A, what is the difference between the lines MetaDisorder, MetaDisorderMD and MetaDisorderMD2?

The MetaDisorder, MetaDisorderMD and MetaDisorderMD2 are three different algorithms (each is a combination of several algorithms) for disorder prediction. We added this explanatory note to the Figure 2A along with the reference with the detailed description.

15. Line 179: May be remove 'and' and start a new sentence?

We removed “and” and started a new sentence (new line 208).

16. Line 190 and Figure 3E: When titrating a C4 competitor in a reaction with a G4 probe, first it eliminates the single stranded G4 by generating a double stranded GC4 probe. Then the excess C4 competes with the GC4 probe on the binding of the protein. Thus the results are consistent with a preference for the single stranded G4 as seen in 3D.

Thank you for pointing this out, we added this interpretation in the Figure 3E legend. This however does not change the conclusion that “HpRif1^NTE^-DNA interaction is weak and lacks clearly defined specificity”, since it poorly differentiates between Tel4(G) and Tel4(GC) (and the non-telomeric dsrnd1 oligo; Figure 3D, E).

17. Line 371: Not in all yeasts the role of Ku is conserved (e.g. K. lactis). I'm not convinced that the binding of Ku to telomerase is conserved and we can assume that it occurs also in H. polymorpha.

We apologize for this mistake. The paragraph is re-written now (lines 449-462). We also added the new result showing that HpKu80 interact with HpTERT in the Y2H assay (Figure 5 —figure supplement 2G). This result may help to explain the telomere shortening in the HpKu mutants and is in agreement with the proposition that transient telomerase binding to telomeres could be mediated via Ku heterodimer.